# Coral photosymbiosis on Mid-Devonian reefs

Jonathan Jung[1 ✉], Simon F. Zoppe[2], Till Söte[3], Simone Moretti[1], Nicolas N. Duprey[1], Alan D. Foreman[1], Tanja Wald[1], Hubert Vonhof[1], Gerald H. Haug[1,4], Daniel M. Sigman[5], Andreas Mulch[2,6], Eberhard Schindler[7], Dorte Janussen[7] & Alfredo Martínez-García[1 ✉]

The ability of stony corals to thrive in the oligotrophic (low-nutrient, low-productivity) surface waters of the tropical ocean is commonly attributed to their symbiotic relationship with photosynthetic dinoflagellates[1,2]. The evolutionary history of this symbiosis might clarify its organismal and environmental roles[3], but its prevalence through time, and across taxa, morphologies and oceanic settings, is currently unclear[4–6]. Here we report measurements of the nitrogen isotope ($^{15}$N/$^{14}$N) ratio of coral-bound organic matter (CB-$\delta^{15}$N) in samples from Mid-Devonian reefs (Givetian, around 385 million years ago), which represent a constraint on the evolution of coral photosymbiosis. Colonial tabulate and fasciculate (dendroid) rugose corals have low CB-$\delta^{15}$N values (2.51 ± 0.97‰) in comparison with co-occurring solitary and (pseudo) colonial (cerioid or phaceloid) rugose corals (5.52 ± 1.63‰). The average of the isotopic difference per deposit (3.01 ± 0.58‰) is statistically indistinguishable from that observed between modern symbiont-barren and symbiont-bearing corals (3.38 ± 1.05‰). On the basis of this evidence, we infer that Mid-Devonian tabulate and some fasciculate (dendroid) rugose corals hosted active photosymbionts, while solitary and some (pseudo)colonial (cerioid or phaceloid) rugose corals did not. The low CB-$\delta^{15}$N values of the Devonian tabulate and fasciculate rugose corals relative to the modern range suggest that Mid-Devonian reefs formed in biogeochemical regimes analogous to the modern oligotrophic subtropical gyres. Widespread oligotrophy during the Devonian may have promoted coral photosymbiosis, the occurrence of which may explain why Devonian reefs were the most productive reef ecosystems of the Phanerozoic.

The Devonian (approximately 419–359 million years ago (Ma)) was a period of higher sea-surface temperatures (23–32 °C)[7–9] and atmospheric carbon dioxide ($CO_2$) (1,000–2,000 ppm)[6,10] than the present. Unlike today, its carbonate chemistry was dominated by calcite precipitation, probably due to lower sea-water magnesium/calcium (Mg/Ca) ratios[11–13]. The Mid-Devonian hosted the most significant expansion of metazoan reefs in the Phanerozoic[6,14], and well-preserved reefs from this period are widespread across present-day Europe, North America, North Africa, Australia, Siberia and China. In the Devonian, these reefs bordered the Rheic Ocean, which lay at the southern margin of Laurussia and northern border of Gondwana[6,15–18] (Fig. 1). Along the southern edge of Laurussia, these ancient reef communities reached their greatest extent and highest diversity during the Givetian stage (around 387–382 Ma)[6,14]. These flourishing metazoan reefs were wiped out diachronically over the course of the Kellwasser Crisis during the late Frasnian (372.2 Ma)[19]. Afterwards, reefs were mainly built by cyanobacteria/algae but were present only in very reduced numbers until the end of the Famennian (the Devonian/Carboniferous boundary)[20–22]. It has been suggested that the ability to host photosymbionts was paramount to the ecological success of ancient reef communities during the Givetian stage[3,6,23] and that the subsequent reef collapse during the Late Devonian was associated with a gradual loss of photosymbiotic associations[2,6,23,24]. However, there is still no clear consensus as to whether photosymbiosis was prevalent in the now-extinct coral groups of the Palaeozoic[3,4,25].

Modern tropical scleractinian coral reefs are home to an intricate symbiotic network of highly diverse organisms[26,27]. Most prominently, an endosymbiotic relationship with single-celled photosynthetic dinoflagellates of the family Symbiodiniaceae allows corals to recycle and retain nutrients and leverage them for organic carbon production, an approach that is particularly strategic in oligotrophic, nutrient-poor surface waters[1,2]. The endosymbiotic algae reside in the gastrodermis of the coral and use the host's metabolic nitrogen (N) waste (ammonium ($NH_4^+$)) for photosynthesis[28–30]. Due to the isotopic fractionation that occurs in de-amination and other metabolic reactions, this ammonium is depleted in $^{15}$N (ref. 28). In symbiont-barren coral species (as in heterotrophic organisms in general), the $^{15}$N-depleted metabolic ammonium is excreted, elevating the $^{15}$N to $^{14}$N ratio (expressed as $\delta^{15}$N = [($^{15}$N/$^{14}$N)$_{sample}$/($^{15}$N/$^{14}$N)$_{air}$ − 1] × 1,000 in ‰) of the coral relative to the food source by 2–4‰ (refs. 29–34). By contrast, in symbiont-bearing corals, metabolic ammonium is translocated from the coral host to the endosymbionts and is thus retained within the coral host–symbiont system[33].

[1]Climate Geochemistry Department, Max Planck Institute for Chemistry, Mainz, Germany. [2]Goethe University Frankfurt, Institute of Geosciences, Frankfurt am Main, Germany. [3]Department of Geology and Paleontology, University of Münster, Münster, Germany. [4]Department of Earth and Planetary Sciences, ETH Zürich, Zurich, Switzerland. [5]Department of Geosciences, Princeton University, Princeton, NJ, USA. [6]Senckenberg Biodiversity and Climate Research Centre, Frankfurt am Main, Germany. [7]Senckenberg Research Institute and Natural History Museum Frankfurt, Frankfurt am Main, Germany. ✉e-mail: jonathan.jung@mpic.de; a.martinez-garcia@mpic.de

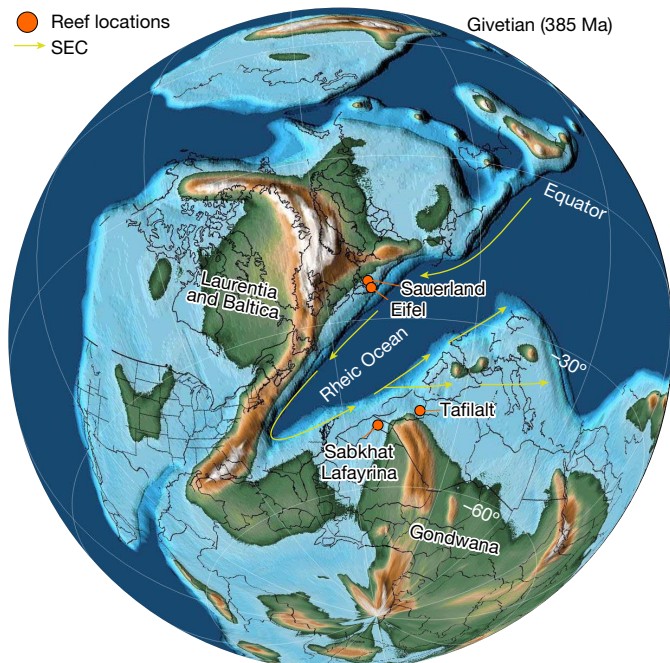

**Fig. 1 | Sample locations relative to a palaeogeographic reconstruction of the continental configuration during the Givetian stage (around 387–382 Ma) of the Devonian period (around 419–359 Ma).** The palaeomap and palaeopositions were generated using GPlates based on the PALEOMAP project of Scotese[86,87]. The South Equatorial Current (SEC) is based on reconstructions and iterations thereof by Dopieralska[15], Jakubowicz et al.[16] and Oczlon[17]. The sampling locations are indicated in orange.

Consequently, the $\delta^{15}N$ of symbiont-bearing corals roughly reflects the $\delta^{15}N$ of the food source without significant isotopic offset[30,35,36]. This $\delta^{15}N$ difference between symbiont-bearing and symbiont-barren corals is reflected in the organic compounds bound within the walls of the coral's mineral skeleton[31,35–37]. Different lines of evidence suggest that the $\delta^{15}N$ of the organic matter bound to the biomineral matrix of corals and other calcifying organisms can remain unaltered for millions of years[31,37–43]. Thus, coral-bound organic matter $\delta^{15}N$ (CB-$\delta^{15}N$) can be used to assess photosymbiotic activity in fossil corals[31,37].

Studies based on the analysis of CB-$\delta^{15}N$ have traced some form of photosymbiosis in fossilized scleractinian corals to the late Triassic (Norian stage, around 212 Ma)[31,37]. In addition, phylogenetic reconstructions based on correlated evolution have placed the emergence of photosymbiosis in corals to the Permian (282.8 ± 16 Ma)[44] and, more broadly, the first photosymbiotic associations with Anthozoa to the Middle Devonian (383 Ma)[45]. Research on ancient photosymbiosis in Palaeozoic corals relies primarily on comparisons with modern Scleractinia, based on morphology[3,4,6,23] or carbonate carbon and oxygen isotopes ($\delta^{13}C$ and $\delta^{18}O$, respectively)[3,23,25,46], which have elicited different interpretations. Photosymbiosis correlates with several dimensions of morphology, such as growth form, corallite size and level of corallite integration. For instance, modern colonial corals are more likely to harbour photosymbionts, whereas solitary corals are less likely to do so[3,47]. Analogously, studies have concluded that Palaeozoic solitary corals were purely symbiont-barren, whereas colonial corals harboured photosymbionts[3,48]. In addition, other studies have found evidence of adaptive morphology in colonial tabulate corals that suggest photosymbiont activity[23,49] reaching as far back as the Silurian (430 Ma)[49]. However, some modern corals provide exceptions. Solitary corals of the genera *Fungia* (Fungiidae) and *Scolymia* (Mussidae) are known to harbour photosymbionts, while colonial corals of the genus *Tubastraea* (Dendrophylliidae) are known to be fully heterotrophic[50]. As a result,

morphological features alone cannot conclusively identify symbiosis across all taxa[51]. Similarly, carbonate $\delta^{13}C$ and $\delta^{18}O$ measurements have been successfully used to distinguish modern symbiont-bearing and symbiont-barren coral species, but they have been deemed inconclusive in their application to Palaeozoic corals due to the potential for diagenetic alteration[31,52,53] and insufficient experimental data on the comparability of $\delta^{13}C$ and $\delta^{18}O$ between calcitic (for example, Palaeozoic) corals and aragonitic (for example, modern scleractinian) corals[54,55].

Here we present analyses of ancient photosymbiosis in Palaeozoic corals using CB-$\delta^{15}N$ in samples from Mid-Devonian reefs. The studied coral samples are from the Givetian stage, from the Hagen-Balve Reef at Binolen (north-western Sauerland), the Eifel region (Sötenich, Dollendorf and Blankenheim synclines) in Germany, the Tafilalt Province in eastern Morocco and the Sabkhat Lafayrina Reef Complex in Western Sahara (Fig. 1). We focused mainly on tabulate corals (pachyporids, alveolitids, roemeriids), various solitary and (pseudo)colonial (dendroid, phaceloid, ceroid) rugose corals, as well as the carbonate sediment matrix in which they were buried (Figs. 2 and 3a and Extended Data Fig. 1). The skeletal architecture and colony integration of rugose corals includes solitary-growth forms, fasciculate pseudocolonial (dendroid and phaceloid architecture) and colonial (cerioid architecture) corals. Tabulate corals are solely colonial and show different calyx architecture (auloporid, pachyporid/ramose, alveolitid). The Palaeozoic CB-$\delta^{15}N$ data were interpreted in the context of CB-$\delta^{15}N$ data from modern pairs of symbiont-bearing coral species (for example, *Porites* spp.) and symbiont-barren coral species (for example, *Tubastraea* spp.) living in the same reef environment and depth, across a range of reef locations characterized by different 'baseline' $\delta^{15}N$ conditions (Fig. 3c and Extended Data Fig. 2). In addition to the CB-$\delta^{15}N$, the coral carbonate $\delta^{18}O$ and $\delta^{13}C$ were measured in all the Mid-Devonian and modern samples (Extended Data Fig. 3).

The analysis of the N isotopic composition of the organic matter bound to the biomineral matrix of the fossil corals was performed on samples that had undergone chemical cleaning, a step that removes organic matter on the surface of the carbonate material, which may have undergone N isotopic alteration by diagenesis and/or may have included exogenous N from natural processes or from contamination while sampling. In general, the uncleaned samples had higher and more variable CB-$\delta^{15}N$ and weight-normalized N contents than the cleaned samples (Methods). For example, a comparison between the cleaned and uncleaned sedimentary matrix samples from each location showed large differences in the mean $\delta^{15}N$ and weight-normalized N contents, as well as in the variance of these measurements (Fig. 2). In addition, the dendroid rugose coral samples had an average weight-normalized N content that was seven times higher in the uncleaned samples than in the cleaned samples, while the tabulate and solitary rugose corals had, on average, two times higher weight-normalized N contents in the uncleaned samples, suggesting significant contamination from exogenous organic matter (Extended Data Fig. 4).

These findings raise concerns about N isotopic reconstructions of low-N environments from the Palaeozoic or earlier that rely on measurements of total sedimentary N or of components of sedimentary N that would have been exposed to the sedimentary environment during deposition or through geological time. Previous attempts to reconstruct changes in the Devonian N cycle have been based on measurements of bulk sediment $\delta^{15}N$ from settings with high deposition rates[56,57]. These studies have suggested a larger range in values and a more negative average value for $\delta^{15}N$ (−3 to 3‰)[56,57] than for the cleaned sedimentary matrix or coral-bound measurements reported here. Even in more recent sediments, bulk sediment $\delta^{15}N$ can be severely altered by diagenesis or contaminated by exogenous N (refs. 58–63). Our uncleaned samples show $\delta^{15}N$ values that are even more variable than those obtained from bulk sediment in previous studies[56,57], illustrating the potential effect of diagenesis and/or contamination with exogenous

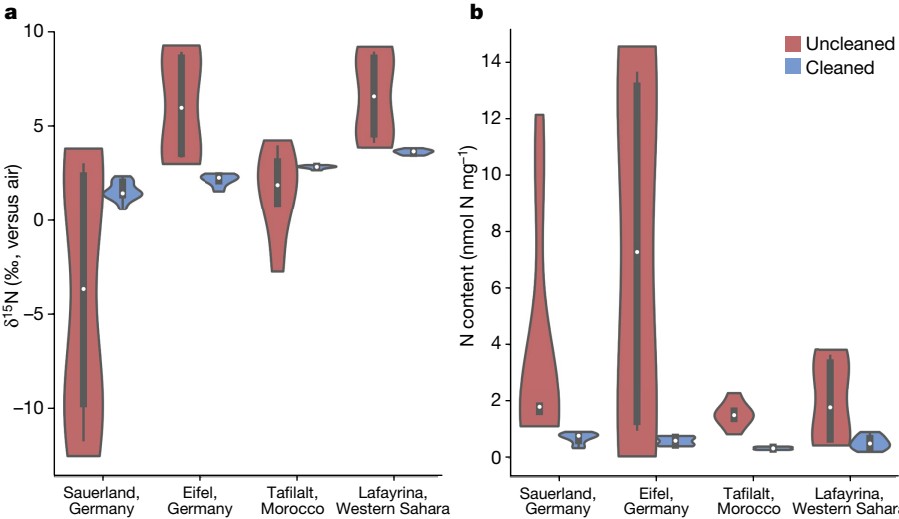

**Fig. 2 | Comparison between cleaned and uncleaned sedimentary matrix samples. a,** Nitrogen isotope values ($\delta^{15}$N in ‰ versus air) of sedimentary matrix material, uncleaned (Sauerland: $n$ = 10, Eifel: $n$ = 6, Tafilalt: $n$ = 6, Lafayrina: $n$ = 6) and cleaned (Sauerland: $n$ = 16, Eifel: $n$ = 6, Tafilalt: $n$ = 6, Lafayrina: $n$ = 6). Cleaning reduced the spread in isotope values for the sedimentary matrix in the Sauerland ($F$ = 837.56, $P$ = 0.03), Eifel ($F$ = 325.16, $P$ = 0.02), Tafilalt ($F$ = 8.54, $P$ = 0.22) and Lafayrina ($F$ = 292.49, $P$ = 0.02) samples. The cleaned samples converged to mean $\delta^{15}$N values of between 0.62 and 3.82‰. **b,** Corresponding weight-normalized N content (in nanomole of N per milligram of powder).

Overall, the N content was very low (less than 1 nmol N mg$^{-1}$) but always higher in the uncleaned samples. Mean values are indicated by the white dots. The lower and upper hinges indicate the first and third quartiles, encapsulating the interquartile range (IQR). The whiskers extend to the smallest and largest values within 1.5 times the IQR from the hinges, depicting the spread of the data. The shape of the violin plot is defined by a kernel density estimate. Statistical significance tests were conducted using either a Welch's $t$-test, given a similar sample size and a heterogeneous variance (indicated by $F \geq 1$), or an individual $t$-test for similar sample sizes and variances (indicated by $F \leq 1$).

N in low-N environments. By contrast, the values converge to a narrow, positive $\delta^{15}$N range in the cleaned samples (Fig. 2 and Extended Data Fig. 4). Therefore, we based our palaeoecological and palaeoenvironmental interpretations on measurements from the cleaned samples, which reflect the fraction of organic matter that was protected by the biomineral matrix.

In the samples from the Hagen-Balve Reef at Binolen, we obtained a mean CB-$\delta^{15}$N of 1.85 ± 0.56‰ ($n$ = 10) from the cleaned tabulate corals (*Roemerolites brevis rhiphaeus*) and 1.45 ± 0.66‰ ($n$ = 13) from the cleaned dendroid rugose corals (*Dendrostella trigemme*) (Fig. 3a). By contrast, the solitary rugose coral samples had a significantly higher CB-$\delta^{15}$N ($P$ < 0.01), with mean values of 5.16 ± 0.88‰ ($n$ = 5) from *Temnophyllum latum*, 5.52 ± 1.49‰ ($n$ = 4) from *Temnophyllum astrictum* and 3.57 ± 0.22‰ ($n$ = 6) from an unidentified rugose coral sample. The samples from the Sötenich Syncline in the Eifel region showed a very similar pattern, with mean CB-$\delta^{15}$N values of 1.64 ± 0.70‰ ($n$ = 10) from the tabulate coral *Roemerolites brevis brevis* and significantly higher $\delta^{15}$N values ($P$ < 0.01) from the solitary rugose coral *Temnophyllum latum* (4.01 ± 0.42‰, $n$ = 7). The cleaned tabulate corals (*Roemerolites brevis brevis*, *Thamnopora cervicornis* and *Thamnopora urensis*) from the Blankenheim Syncline had mean CB-$\delta^{15}$N values of 2.84 ± 0.18‰ ($n$ = 3), 3.12 ± 0.30‰ ($n$ = 9) and 3.66 ± 0.26‰ ($n$ = 3), respectively, whereas the cerioid rugose coral *Argutastraea quadrigemina* had significantly higher CB-$\delta^{15}$N values of 5.94 ± 0.40‰ ($n$ = 3) ($P$ < 0.01). Similarly, the cleaned tabulate corals (*Alveolites intermixtus intermixtus* and *Alveolites intermixtus minor*) from Tafilalt, Morocco, had mean CB-$\delta^{15}$N values of 3.06 ± 0.44‰ ($n$ = 3) and 3.46 ± 0.49‰ ($n$ = 3), respectively, while the cleaned solitary rugose samples from the same location had significantly higher mean CB-$\delta^{15}$N values of 6.03 ± 0.86‰ ($n$ = 3) from *Siphonophrentis* sp., 6.03 ± 0.12‰ ($n$ = 3) for *Mesophyllum* (*Mesophyllum*) cf. *lissingenense*, and 5.87 ± 0.12‰ ($n$ = 3) for *Acanthophyllum concavum* ($P$ < 0.01). The highest CB-$\delta^{15}$N values from the cleaned tabulate corals were obtained from Sabkhat Lafayrina, Western Sahara, these being 3.48 ± 0.10‰ ($n$ = 3) from *Thamnopora angusta* and 4.13 ± 0.15‰ ($n$ = 3) from *Scoliopora*? sp. The cleaned rugose coral samples from Sabkhat Lafayrina had consistently higher CB-$\delta^{15}$N values of 7.69 ± 0.12‰

($n$ = 3), 6.78 ± 0.42‰ ($n$ = 3) and 7.34 ± 0.23‰ ($n$ = 3) from *Disphyllum?* sp., *Mesophyllum* (*Cystiphylloides*) *secundum* and *Acanthophyllum concavum*, respectively.

The CB-$\delta^{15}$N values obtained from different individuals of the colonial species *Romerolites brevis rhiphaeus* (1.85 ± 0.56‰) and *Dendrostella trigemme* (1.45 ± 0.66‰) from the Hagen-Balve Reef are statistically indistinguishable from those from the *Romerolites brevis brevis* samples from the Sötenich Syncline (1.64 ± 0.70‰). Similarly, the average CB-$\delta^{15}$N value of multiple species of solitary rugose corals from the Hagen-Balve Reef (4.62 ± 1.24‰) was close to those from the Dollendorf (3.06 ± 0.47‰), Sötenich (4.01 ± 0.42‰) and Blankenheim (5.94 ± 0.40‰) synclines in the Eifel region. At the same time, the CB-$\delta^{15}$N values of the tabulate and rugose coral samples from the Blankenheim Syncline are indistinguishable from those from Tafilalt and Lafayrina.

The average difference in CB-$\delta^{15}$N of the cerioid, phaceloid and solitary rugose corals compared to that of the tabulate and dendroid rugose corals ($\Delta^{15}$N$_{\text{CS-CD}}$) was statistically significant ($P$ < 0.01) and remarkably similar between samples from the Sauerland ($\Delta^{15}$N$_{\text{CS-CD}}$ = 2.92 ± 0.98‰), the Eifel region ($\Delta^{15}$N$_{\text{CS-CD}}$ = 2.74 ± 0.29‰), Tafilalt ($\Delta^{15}$N$_{\text{CS-CD}}$ = 2.90 ± 0.25‰) and Sabkhat Lafayrina ($\Delta^{15}$N$_{\text{CS-CD}}$ = 3.50 ± 0.60‰). These isotopic differences are also similar to those observed between modern symbiont-barren and symbiont-bearing corals ($\Delta^{15}$N$_{\text{BA-BE}}$) living in comparable reef environments (average $\Delta^{15}$N$_{\text{BA-BE}}$ = 3.38 ± 1.05‰) (Fig. 3b,d and Extended Data Fig. 6). Our modern dataset demonstrates that the isotopic difference between symbiont-barren and symbiont-bearing corals is consistent across reef systems characterized by different baseline $\delta^{15}$N values for their nitrate supply (Fig. 3c and Extended Data Fig. 2). The lowest average $\Delta^{15}$N$_{\text{BA-BE}}$ values were found in corals from Socotra ($\Delta^{15}$N$_{\text{BA-BE}}$ = 1.97‰) and Cape Verde ($\Delta^{15}$N$_{\text{BA-BE}}$ = 2.35‰), the highest average $\Delta^{15}$N$_{\text{BA-BE}}$ was from Hong Kong ($\Delta^{15}$N$_{\text{BA-BE}}$ = 5.02‰), while corals from Jamaica ($\Delta^{15}$N$_{\text{BA-BE}}$ = 4.26‰), Colombia ($\Delta^{15}$N$_{\text{BA-BE}}$ = 3.14‰) and Brazil ($\Delta^{15}$N$_{\text{BA-BE}}$ = 3.51‰) had values closer to the mean $\Delta^{15}$N$_{\text{BA-BE}}$ (Extended Data Fig. 6). The differences observed in the magnitude of the species offsets may relate to the efficiency of nutrient recycling by

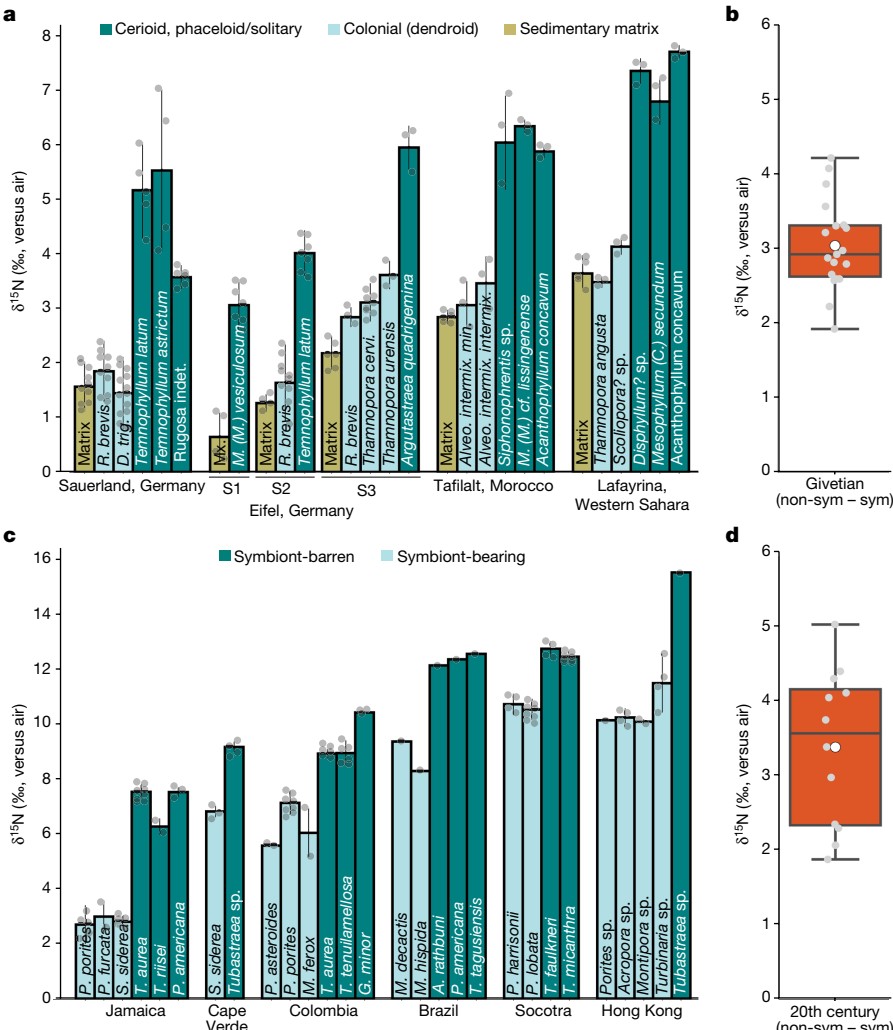

**Fig. 3 | Nitrogen isotope values of Palaeozoic and modern corals. a**, Cleaned CB-$\delta^{15}$N values (in ‰ versus air) of the sedimentary matrix, tabulate corals, dendroid rugose corals and solitary rugose corals from the Hagen-Balve Reef in Binolen, of the sedimentary matrix and a solitary rugose coral from the Dollendorf Syncline (S1), of the sedimentary matrix, tabulate corals and a solitary rugose coral from the Sötenich Syncline (S2), and of the sedimentary matrix, tabulate corals and a cerioid rugose coral from the Blankenheim Syncline (S3) of the Eifel region from the southern edge of Laurussia, bordering the Rheic Ocean. The sedimentary matrix, tabulate corals and solitary rugose corals from the present-day Tafilalt region of Morocco and the sedimentary matrix, tabulate corals, a phaceloid rugose coral and solitary rugose corals from Sabkhat Lafayrina, Western Sahara are from northern Gondwana. Several measurements of the same species were taken together, and the respective intraspecific variation (±1 standard deviation (s.d.)) is shown by vertical lines.

**b**, Average isotopic differences (expressed as $\Delta\delta^{15}$N = $\delta^{15}$N$_{\text{non-sym./solitary/ceroid}}$ − $\delta^{15}$N$_{\text{sym./colonial}}$) between the solitary and colonial species ($n$ = 18). The white dot represents the average value, while the middle line represents the median value. The lower and upper bounds of the box correspond to the first and third quartiles. The upper whisker extends from the upper bound of the box to the largest value within 1.5 times the IQR from the hinge, while the lower whisker extends from the lower bound of the box to the smallest value within 1.5 times the IQR from the hinge. Values beyond the whiskers are considered outliers and are plotted individually. **c**, $\delta^{15}$N of symbiont-bearing and symbiont-barren species from Jamaica, Cabo Verde, the Caribbean side of Colombia, Brazil and Hong Kong. All corals were taken from the same reef depth and are the same age. **d**, Average difference between the symbiont-barren (non-sym.) and symbiont-bearing (sym.) species from all locations ($n$ = 12). Mx., matrix.

coral symbionts, the feeding behaviour of the corals or the degree of nitrate assimilation by coral symbionts[34–37,64]. In any case, the $\Delta\delta^{15}$N$_{\text{BA-BE}}$ values were consistent in all cases with an expectation of the retention of a significant part of the metabolic ammonium within the host–symbiont system of the symbiont-bearing corals, in further support of $\Delta\delta^{15}$N$_{\text{BA-BE}}$ as an indicator of the presence/absence of coral photosymbionts[30,31,37,65].

The average $\Delta\delta^{15}$N$_{\text{CS-CD}}$ observed in the Mid-Devonian samples from the Sauerland, the Eifel synclines, Tafilalt and Sabkhat Lafayrina (average $\Delta\delta^{15}$N$_{\text{CS-CD}}$ = 3.01 ± 0.58‰) is statistically indistinguishable ($F$ = 0.01, $P$ = 0.27, Welch's $t$-test) from the $\Delta\delta^{15}$N$_{\text{BA-BE}}$ observed in modern corals (average $\Delta\delta^{15}$N$_{\text{BA-BE}}$ = 3.38‰ ± 1.05‰) (Fig. 3b,d and Extended Data Fig. 6b). Thus, our CB-$\delta^{15}$N measurements indicate that tabulate and

fasciculate (dendroid) rugose corals hosted active photosymbionts, whereas solitary rugose corals and some rugose corals with fasciculate (phaceloid) morphology and higher colony integration (cerioid architecture) did not. This is thus the oldest conclusive geochemical expression of the presence and absence of photosymbiotic associations in corals to date, and it suggests that autotrophic and heterotrophic corals co-existed on extinct reefs much as they do today.

Variation in absolute CB-$\delta^{15}$N values across sites is to be expected, given the potential for spatial variation in the $\delta^{15}$N of the N supply to reefs[34]. However, the average CB-$\delta^{15}$N difference between the cerioid, phaceloid and solitary rugose corals and the tabulate and dendroid rugose corals ($\Delta\delta^{15}$N$_{\text{CS-CD}}$) was remarkably consistent across sites (Fig. 3a). This is an important finding, given that the sites experienced

very different diagenetic histories. The conodont colour alteration index (CAI) indicates that samples from the Hagen-Balve Reef experienced maximum temperatures of 190–300 °C (Supplementary Table 1) and Tafilalt experienced maximum temperatures of 155–230 °C (refs. 15,66) while the temperatures experienced by samples from the Eifel region did not exceed 50–95 °C (ref. 67). The similarity in the CB-$\delta^{15}$N from these locations is consistent with results from laboratory heating experiments, which have shown no significant changes in CB-$\delta^{15}$N despite significant decreases in the weight-normalized N content at temperatures of 300 °C (ref. 41), suggesting that alteration-driven exposure and the subsequent loss of previously protected N does not significantly affect the N isotopic composition of the remaining coral-bound organic matter. The CB-$\delta^{15}$N values of our Sauerland, Eifel, Moroccan and Western Saharan samples showed no correlation with the N contents, further supporting this interpretation (Extended Data Fig. 5). The consistency of the $\Delta\delta^{15}N_{CS-CD}$ values across sites, as well as the lack of correlation between the weight-normalized N content and CB-$\delta^{15}$N, strongly suggest that the measured coral-bound organic matter is indeed native to the organisms and has not been isotopically altered by its long residence in the geological record and the wide range of temperatures experienced by the fossils.

Interestingly, the $\delta^{15}$N values from the cleaned sedimentary matrix from Binolen (1.57 ± 0.46‰, $n$ = 16), the Dollendorf Syncline (0.83 ± 0.13‰, $n$ = 4), Sötenich Syncline (1.27 ± 0.16‰, $n$ = 4), Blankenheim Syncline (2.18 ± 0.31‰, $n$ = 6), Tafilalt (2.84 ± 0.08‰, $n$ = 6) and Sabkhat Lafayrina (3.64 ± 0.13‰, $n$ = 6) were all similar to the CB-$\delta^{15}$N values of the tabulate or dendroid rugose coral samples from their respective deposits (Fig. 3a). The sedimentary matrix consisted mainly of fine bioclastic debris with abundant micrite. This bioclastic debris was probably dominantly sourced from the major calcifiers, including the tabulate and dendroid rugose corals[68], which is consistent with their photosymbiosis increasing their growth rate. Thus, the isotopic similarity of the sedimentary matrix and the colonial corals may simply reflect that the matrix is largely composed of the remains of these corals. Our findings raise the possibility that, unlike bulk sediment measurements, the analysis of the biomineral-bound N isotopic composition of sedimentary rocks rich in biogenic carbonate might provide information about past changes in the N cycle even when they do not contain recognizable macrofossils, provided that the surficial organic N on the biomineral grains is removed by chemical cleaning. If confirmed, this type of measurement would provide a new lens through which to investigate changes in the N cycle across broad ranges of geological time and palaeoenvironments.

The low average value for CB-$\delta^{15}$N reported here may offer insights into ocean N cycling during the Mid-Devonian. The $\delta^{15}$N in corals is sensitive to the $\delta^{15}$N of the fixed N supplied to their oligotrophic reef environment, which is typically dominated by the nitrate supplied from the shallow subsurface by mixing and/or upwelling[35,69], with exceptions in coastal systems with large terrestrial (including anthropogenic) N sources[64,70–72]. Accordingly, the large range in CB-$\delta^{15}$N values across our modern sampling sites can be attributed to distinct processes in the marine N cycle that affect the $\delta^{15}$N of the N supplied to each reef (Extended Data Fig. 2). The CB-$\delta^{15}$N values were lowest from Jamaica (CB-$\delta^{15}$N = 2.87 ± 0.28‰), which is located in the central Caribbean. In this region, the $\delta^{15}$N of the nitrate supply to the euphotic zone is low, largely due to regional $N_2$ fixation and its remineralization to low-$\delta^{15}$N nitrate in the thermocline[73,74]. By contrast, the highest values were obtained from two nutrient-rich systems—Socotra (CB-$\delta^{15}$N = 10.59 ± 0.38‰) and Hong Kong (CB-$\delta^{15}$N = 10.66 ± 0.90‰) (Fig. 3c). Socotra is located in the vicinity of one of the largest oceanic oxygen-deficient zones, with high rates of water-column denitrification—a process that thus elevates the $\delta^{15}$N of the subsurface nitrate that is supplied to the surface[75]. The estuary outside of Hong Kong, in contrast, is influenced by anthropogenic activities in the Pearl River Basin that tend to elevate the $\delta^{15}$N of both the ammonium and

nitrate sources (for example, ammonium oxidation coupled to denitrification)[64,65]. The CB-$\delta^{15}$N from Colombia (CB-$\delta^{15}$N = 5.96 ± 1.51‰) and Cape Verde (CB-$\delta^{15}$N = 6.83 ± 0.20‰) had intermediate values characteristic of the mean ocean pycnocline nitrate[76,77].

The Devonian mean CB-$\delta^{15}$N values from colonial corals from the initial Hagen-Balve Reef at Binolen in Sauerland (1.53 ± 0.58‰), Sötenich (1.64 ± 0.70‰), the Blankenheim Syncline (3.12 ± 0.37‰) of the Eifel region, Tafilalt in Morocco (3.26 ± 0.28‰) and Sabkhat Lafayrina in Western Sahara (3.81 ± 0.46‰) are similar to those found in the western tropical and subtropical North Atlantic[36,78], a region dominated by strong density stratification, surface nutrient depletion and low surface chlorophyll concentrations. The low $\delta^{15}$N of the thermocline nitrate in this region and similar nitrate isotopic features in other subtropical gyres[79,80] probably derive from $N_2$ fixation, which is largely restricted to N-deplete surface waters[81] and which lowers the thermocline nitrate $\delta^{15}$N most strongly in the low-nutrient subtropical gyres[74]. Thus, the low CB-$\delta^{15}$N we observed in each of the fossil reefs may indicate that they occurred in nutrient-poor environments associated with a westward-intensified subtropical gyre. This supports the view that the reefs of the Givetian (around 385 Ma), which comprised some of the most widespread and diverse reef biotas of the Phanerozoic, were adapted to nutrient-poor conditions, as applies broadly to the symbiont-bearing scleractinian coral reefs of today[6,20–22]. Thus, the success of symbiotic corals in the Givetian may have been linked to the occurrence of extensive coastal regions under the influence of strongly stratified, nutrient-poor conditions that characterized the western ocean margins at tropical and subtropical latitudes. The CB-$\delta^{15}$N range across Givetian deposits is in the low end of the range observed in the modern ocean[34] (Fig. 3b). The lowest $\delta^{15}$N values were recorded from sites occurring at lower latitudes and on the western margin of the small gyre reconstructed from the Rheic Ocean, consistent with the region with the lowest nitrate $\delta^{15}$N observed in modern subtropical gyres[73,74].

The Givetian coral CB-$\delta^{15}$N values from Sauerland were lower than those measured from any modern coral specimen. While this observation may simply be an artefact of the limited number of sites, it may also reflect characteristics of the Givetian Ocean, in which case, there are several possible explanations for it. First, it may reflect natural environmental isotopic gradients. For example, the Givetian may have been characterized by an intensification of the low-$\delta^{15}$N features associated with tropical and subtropical waters. This might have occurred if the $N_2$ fixation rates were greater and/or if the subtropical gyres were more expansive and characterized by a deeper thermocline. Subtropical gyre expansion may have been driven by the warm climate of the Givetian, consistent with climate model experiments of warming in which the atmospheric Hadley cells expand[82,83]. A particularly deep western thermocline may also have been encouraged by the very wide ocean basin of the Givetian (Fig. 1). Alternatively, the low CB-$\delta^{15}$N of the Givetian warm period may reflect a reduction in the importance of water-column denitrification in oceanic N loss[84], such as would be associated with a contraction of ocean suboxic zones. This would be consistent with observations of minimal water-column denitrification during warm periods of the Cenozoic, which indicate that ocean suboxia is reduced under warmer climates[38,40,43].

These early signals of photosymbiosis in corals from the Mid-Devonian indicate that it supported coral productivity under warm climatic conditions. The late Triassic and early Miocene—subsequent periods during which coral photosymbiosis has been reconstructed using nitrogen isotopes[31,42]—were also warmer than today. By contrast, under modern global warming due to anthropogenic greenhouse gas emissions, coral bleaching and associated mass mortality events point to a warming-driven breakdown of their symbiosis as perhaps being the greatest threat to the future of scleractinian coral reefs[85]. The robustness of coral photosymbiosis during past warm climates indicates that the failure of coral symbiosis under ongoing global warming is not due to the elevated surface-ocean temperatures being

reached, but rather the rapidity with which surface-ocean temperatures are rising, which may be outstripping the ability of the symbiotic relationship to adapt.

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

## Methods

### Geological setting and stratigraphy

The main material was collected near a cliff at the top of the Binolen section (the 'C-layers' after Löw et al.[68]; GPS coordinates 51° 22′ 12″ N, 7° 51′ 27″ E) in the Hönne Valley in north-western Sauerland, Germany. The Binolen section is located in the northern Rhenish Massif at the eastern edge of the Remscheid–Altena Anticline, which is surrounded by carbonate platform deposits of the Hagen-Balve Reef. In terms of stratigraphy, the base of the Binolen section lies in the lower Givetian (probably within the *timorensis* Conodont Zone), defining the lower boundary of the basal part of the Hagen-Balve Formation (Binolen Member)[68]. However, the cliff at the top of the Binolen Member falls within the lower/middle Givetian boundary interval[68,88].

During the Givetian, the Hagen-Balve Reef developed as an elongated carbonate platform surrounding a local submarine high on the Rhenish shelf, at the southern tip of Laurussia (Fig. 1). The onset of reef formation was approximately isochronous in the early Givetian[68]. The depositional history of the initial reef formation of the Binolen Member has been divided into several depophases[68]. The samples analysed in this study were collected from strata in the upper part of Depophase VI (Beds 59 to 65 of the C-layers)[68] and stem from the initial reef platform of the Hagen-Balve Reef. This part of the initial reef formation of the Binolen Member is characterized by coral–stromatoporoid frame rudstones and coral–stromatoporoid-dominated float-bafflestones, representing a semi-open carbonate platform with argillaceous sediment input[68].

Samples from the Eifel region were provided by the Senckenberg Research Institute and Natural History Museum Frankfurt. The limestone synclines of the Eifel region are located between the Lower Rhine Bay to the north and Trier Bay to the south. Geologically, the region is part of the Rhenish Massif and consists of Devonian slates, sandstones and limestones interspersed with bioclasts, which were deposited in a coastal setting south of Laurussia[89,90] (Fig. 1).

The Sötenich Syncline is characterized by changing assemblages of thinly bedded marly mudstones and thick layers of gastropod–coral–trilobite wackestones to floatstones, which merge into stromatoporoid–coral rudstones in the uppermost section. The coral associations are indicative of a low-energy regime in a shallow-marine lagoon. The faunal composition and facies types in the upper section suggest elevated sedimentary input and elevated nutrient supply[90].

The Dollendorf Syncline has yielded a rich macrofauna characteristic of the Mid-Devonian. The local limestones are mainly composed of calcisphere–ostracod wackestones or packestones and amphiporoid floatstones, indicating a shallow-marine lagoonal setting with restricted, low-energy water flow. Interspersed amphiporoid rudstones suggest periods with high-energy regimes, potentially more influenced by open-marine conditions[89].

The Blankenheim Syncline is dominated by Mid-Devonian carbonate platform facies and biostromal reef deposits. Siltstones and mudstones are occasionally interbedded as the clay content increases towards the eastern part of the syncline, consistent with a marginal reef setting with a partially open-ocean influence[91].

The Mid-Devonian outcrops of the Tafilalt Platform (GPS coordinates 31° 20′ N, −4° 16′ W) are characterized by shallow to pelagic ridge topographies with very low sedimentation rates. The fossil-rich deposits are predominantly of shallow-water origin, close to an inclined carbonate ramp[92].

The Sabkhat Lafayrina Reef is located on the southern edge of the Tindouf Basin in Western Sahara (GPS coordinates 26° 33′ 04″ N, 11° 29′ 32″ W). The reef consisted of siliciclastic shoals with enveloping reefal carbonates. The benthic assemblages are reworked, but all autochthonous[93].

### Thin-section analyses and sample storage

To taxonomically identify the collected coral samples from Binolen, fossil-rich rock samples were cut systematically to produce longitudinal and cross-sections of individual coral skeletons. Thin sections were prepared with a thickness of 70–80 μm. Microphotographs were taken under transmitted light using a Keyence VHX-6000 digital microscope to identify the tabulate and rugose corals based on refs. 90,94–96 (and references therein).

Nine thin sections from the initial Hagen-Balve Reef at Binolen will be stored at the Geomuseum of the Westfälische Wilhelms University in Münster (GMM) under the inventory numbers GMM B2C.59-1 to GMM B2C.59-9 (Supplementary Fig. 1).

The Eifel, Moroccan (Tafilalt) and Western Saharan (Sabkhat Lafayrina) samples were provided by the Senckenberg Research Institute and Natural History Museum Frankfurt, Germany, and included *Roemerolites brevis brevis* (SMF 40159) and *Temnophyllum* cf. *ornatum* (= *T. latum*) (SMF 40367/2) from the Sötenich Syncline; *Mesophyllum* (*Mesophyllum*) *vesiculosum* (SMF 73856) from the Dollendorf Syncline; and *Roemerolites brevis brevis* and *Argutastraea quadrigemina* (SMF 40160), *Thamnopora cervicornis* (1) (SMF 40256), *Thamnopora cervicornis* (2) (SMF 40255) and *Thamnopora urensis* (SMF 40213) from the Blankenheim Syncline. The Moroccan (Tafilalt) samples included *Mesophyllum* (*Mesophyllum*) cf. *lissingenense* (SMF 75853), *Acanthophyllum concavum* (SMF 75854), *Siphonophrentis* sp. (SMF 75855), *Alveolites intermixtus intermixtus* (SMF 75856) and *Alveolites intermixtus minor* (SMF 75857). The Western Saharan samples were collected from the same locality in Sabkhat Lafayrina and included *Mesophyllum* (*Cystiphylloides*) *secundum* (SMF 99529), *Acanthophyllum concavum* (SMF 99530), *Thamnopora angusta* (SMF 99531), *Scoliopora*? sp. (SMF 995302) and *Dispyllum*? sp. (SMF 70205) (Supplementary Fig. 2).

### Conodont colour alteration index

Assessing the textural alteration of conodonts has been used for some time as a proxy for the maturation of rocks[97], with the first systematic approach to quantifying the temperature regimes experienced by a rock during diagenesis using the CAI[98,99]. Generally, conodont elements are composed of calcium phosphate (frankolite)[97]. During the growing phase of the conodont animal, frankolite lamellae are separated by thin organic layers. This organic matter can alter as a consequence of a carbonization reaction, changing colour in a characteristic way, this being the basis of the CAI (Supplementary Fig. 3 and Supplementary Table 1). Since then, several authors have successfully used the CAI to assess and quantify the maturation of regional rock formations and basins[100–106].

The CAI has been used in the Rhenish Massif[103,107]. Helsen and Königshof[103] produced a useful map of CAI isoclines for the region. We used 30 conodonts from Binolen to determine the temperature-induced diagenetic overprint of the limestones collected from slightly older strata only a few metres away, and narrowed the values down to 4.0–4.5 (corresponding to maximum temperatures of 190–300 °C)[108] for most of the Mid-Devonian strata of the Rhenish Massif. The CAIs of the different synclines of the Eifel Hills yielded nearly homogeneous values of between 1.5 and 2.0 (corresponding to maximum temperatures of 50–95 °C)[67,103,109]. The CAI values for our samples from the Tafilalt Platform in Morocco and the outskirts of the Anti-Atlas were generally between 3.5 and 4 (corresponding to maximum temperatures of 155–230 °C)[15,66,107,108].

### Analysis of coral-bound nitrogen isotopes

The CB-δ[15]N measurements were performed in the Martínez-García Laboratory at the Max Planck Institute for Chemistry in Mainz. We used the persulfate oxidation–denitrifier method[78,110], first applied to corals by Wang et al.[34,36], with the analytical modifications described by Moretti et al.[111].

The collected samples of fossil-rich carbonate rocks were cut into smaller hand specimens using a stationary rock saw. Sample material was carefully extracted from these using a millimetre drill bit attached to a hand-held Dremel. Only specimens sampled from the edge of a hand piece were considered to ensure that the different phases of material (coral skeleton, secondary sparite and surrounding carbonate sediment) and their respective dimensions were visible (Extended Data Fig. 1). Each phase was collected exclusively from the centre of the mass to minimize the contamination of adjacent material (Extended Data Fig. 1). Subsequent samples were sieved to separate coarse (250–63 µm) and fine (63–5 µm) aliquots. The coarse fraction was used for [15]N-isotope analysis, while the fine fraction was further prepared for [13]C and [18]O analysis.

First, $20 \pm 2$ mg of uncleaned, coarse powder was weighed into a 12 ml tube. Subsequently, to remove the clay fraction, 10 ml of a 2% sodium polyphosphate solution was added. This mixture was left on a shaker at 120 rpm for 5 min and then placed in an ultrasonic bath for 1 min. After this, the tubes were taken out and the supernatant was decanted. Then, 8–10 ml of Milli-Q water was added and the samples were centrifuged at 300 rpm for 2 min before being removed. The procedure was repeated three times.

To remove potential iron-manganese oxides, 5 ml of pH-adjusted dithionite–citric acid (pH 8) was added to each sample tube, which was placed in an 80 °C deionized-water bath for 30–40 min. The samples were removed and centrifuged, the supernatant was decanted, and the sample was rinsed three times with Milli-Q (see steps above). Afterwards, sample material was transferred to a previously muffled 4 ml VWR borosilicate glass vial and 3 ml of a potassium peroxydisulfate oxidative reactant (POR) solution (pH > 12) was added. The samples were then autoclaved at 121 °C for 65 min for the oxidation of non-bound organic matter. Finally, the supernatant was removed using a muffled pipette attached to a vacuum line set at 500 mbar, and the sample was rinsed at least three times with Milli-Q. The cleaned samples were stored in a drying oven at 60 °C overnight.

Once the powder had fully dried, $15 \pm 5$ mg of cleaned powder was weighed inside a clean room to minimize contamination. Thereafter, skeletal organic matter was released by dissolving the cleaned powder with 4 N hydrochloric acid (HCl). This led to a solution of calcium chloride ($CaCl_2$) at a pH of less than 2. The amount of 4 N HCl used was calculated on the basis of the sample weight. We used the stoichiometric calculation of the reaction ($CaCO_3 + 2HCl \rightleftharpoons CaCl_2 + H_2O + CO_2$), which translated to 5 µl 4 N HCl per 1 mg of cleaned carbonate powder. We added an additional 20 µl 4 N HCl to each sample to ensure complete dissolution.

Concurrently, a new POR solution was prepared inside the clean room with 0.7 g of potassium peroxydisulfate and 4 ml of 6.25 N sodium hydroxide (NaOH), filled to 100 ml with Milli-Q water. Then, 1 ml of POR solution was pipetted onto each dissolved sample and into at least 10 empty cleaned vials (blanks), and the batch of vials was placed in a custom-built sample rack that was tightly sealed with a polytetrafluoroethylene sheet before being autoclaved at 121 °C for 65 min. After the autoclave run, the supernatant was tested for its pH to make sure every sample was basic (pH > 10). Eventually, each sample was balanced with the same aliquot of HCl previously used for dissolution so as to achieve a pH close to 7. From the resulting solution, the nitrate concentration was measured for each sample by quantitative conversion to nitric oxide and subsequent chemiluminescence detection[112].

A volume of 1 ml of concentrated denitrifying bacteria (*Pseudomonas chlororaphis*) was injected into 800 ml of growth media and left for 4–6 days to grow in the dark at room temperature on a shaking rack. Once the bacteria had grown sufficiently, the medium was transferred to autoclaved polyethylene bottles and centrifuged at 7,600 rpm for 10 min. The supernatant was then discarded and the remaining bacterial pellet was resuspended using a buffered (pH 6.3) resuspension medium. From this, 3 ml were pipetted into muffled 12 ml glass vials, which were capped with a septum, tightly sealed, and placed upside-down on a needle rack with a small extra needle for pressure release. The needle rack supplied a continuous flow of $N_2$ for at least 3 h to replace the internal atmosphere with pure $N_2$. The vials of bacteria were removed from the rack, and approximately 0.8 ml of the oxidized sample was injected into each vial. Once all the samples had been injected, the bacterial vials were placed in the dark for 2–3 h to ensure the quantitative transformation of nitrate to nitrous oxide before being frozen at −21 °C.

On the day of the analysis, the bacteria were thawed, lysed with several drops of 10 N NaOH and finally placed in a mass spectrometer for isotopic analysis. The $\delta^{15}N$ of the $N_2O$ was determined by a purpose-built inlet system coupled to a Thermo MAT253 Plus stable isotope ratio mass spectrometer[110,113]. Long-term precision was determined by running internal carbonate standards with each sample batch, which yielded an average carbonate standard reproducibility of ±0.2‰. The average reproducibility for the replicate Devonian coral measurements was 0.22‰ ($n = 45$) and 0.68‰ ($n = 20$) for the cleaned and uncleaned samples, respectively.

The modern samples of *Tubastraea* spp. and *Porites* spp. from Cape Verde, Colombia, Jamaica and Socotra were taken from four different collections held in the Senckenberg Research Institute and Natural History Museum Frankfurt. Subsequent samples were drilled with a hand-held Dremel, and the powder was transferred into 4 ml borosilicate glasses using aluminium foil. Each sample was then sieved into coarse (250–63 µm) and fine (63–5 µm) fractions, with 6 mg coarse and 100–200 µg fine powder being used for the $\delta^{15}N$, $\delta^{13}C$ and $\delta^{18}O$ analyses, respectively.

For analysis of the modern coral samples, 8 mg of cleaned coarse powder was weighed into a 4 ml VWR borosilicate glass vial and filled with 4.25 ml of 2% sodium hypochlorite before being left on a shaking table at 120 rpm for at least 24 h. Afterwards, the supernatant was removed using a muffled pipette attached to a vacuum line set at 500 mbar and was further treated as described for the Palaeozoic samples.

### Coral oxygen and carbon isotopes

Amounts of 100–200 µg of coral carbonate sample material were analysed for $\delta^{18}O$ in the inorganic stable isotope laboratory at the Max Planck Institute for Chemistry in Mainz. In a run of 55 samples, one International Atomic Energy Agency carbonate standard (IAEA-603) ($n = 10$) and one Virje University Internal Carbonate Standard (VICS) ($n = 11$) were used to calibrate the analyses to the Vienna Pee Dee Belemnite scale. The samples were analysed using an isotope ratio mass spectrometer (IRMS) (Delta V Advantage, Thermo Scientific) connected to a GasBench II unit (Thermo Scientific). Each sample was placed in a 12 ml Exetainer vial (part no. 9RK8W, Labco). The samples and standards were then put into a 70 °C-heated hot block. First, the vials were flushed with helium (He) to remove the atmospheric $CO_2$. Then, 5–10 drops of more than 99% phosphoric acid ($H_3PO_4$) were added and the sample was left to dissolve for 1.5 h. Finally, the sample was transferred in He carrier gas to the GasBench II unit, where water and contaminant gases were removed before subsequent isotope analysis in the IRMS. The average analytical precision, based on the reproducibility of IAEA-603, was 0.11‰ (1 s.d., $n = 42$) and 0.09‰ (1 s.d., $n = 42$) for $\delta^{18}O$ and $\delta^{13}C$, respectively.

The Palaeozoic samples showed mean $\delta^{18}O$ values ranging from −3.98‰ for the cerioid rugose coral samples to −7.44‰ for the secondary sparite samples (Supplementary Table 4). The mean $\delta^{13}C$ values for all the samples clustered around 1.72‰, with the lowest $\delta^{13}C$ values recorded for solitary rugose corals (−0.98‰). Notably, all Givetian coral samples clustered within narrow $\delta^{18}O$ and $\delta^{13}C$ values (−3.98 to −7.40‰ and 1.52 to −0.98‰, respectively).

The skeletal $\delta^{18}O$ and $\delta^{13}C$ values from the modern samples were relatively widespread, ranging from −7.39‰ to 3.57‰ and −10.27‰ to 1.60‰, respectively (Supplementary Table 5). Symbiont-bearing and symbiont-barren coral species did not show any distinct offset

in either $\delta^{18}O$ or $\delta^{13}C$. However, the modern symbiont-bearing and symbiont-barren species were distinguishable from a cross-plot of the $\delta^{18}O$ versus $\delta^{13}C$ values[46] (Extended Data Fig. 3).

The original $\delta^{18}O$ and $\delta^{13}C$ of corals can be altered by the partial dissolution of aragonite, the precipitation of secondary carbonates or the recrystallization of metastable aragonite to calcite[114–117]. While secondary carbonates (sparite) are predominantly observed in submarine environments[116], partial dissolution or recrystallization are more common in subaerial settings[114,118]. According to previous studies on the Hagen-Balve Reef and the Eifel region, the samples have probably been subjected to both submarine and subaerial alteration[68,89,90].

The $\delta^{18}O$ and $\delta^{13}C$ values from our samples from Binolen and the Eifel region clustered within the ranges previously discussed for marine limestones[119]. The narrow ranges of the $\delta^{18}O$ and $\delta^{13}C$ values suggest photosymbiosis across tabulate and rugose coral species[25,46,120], thus standing in contrast to the distinctions identified from the CB-$\delta^{15}N$ measurements (Fig. 3a). Previous studies have highlighted that diagenetic processes and geochemical comparisons of polymorphs (that is, calcitic skeletons for Palaeozoic coral samples and aragonitic skeletons for modern scleractinians) can bias the interpretation of $\delta^{18}O$ and $\delta^{13}C$ values and thus are thought to be less robust proxies for fossil reef settings[53,115,118]. In addition, increasing temperatures and recrystallization can bias carbonate samples towards more negative $\delta^{18}O$ values[121,122]. Thus, it is possible that the diagenetic alteration of coral carbonate $\delta^{18}O$ can bias interpretations towards symbiotic associations.

## Statistics and reproducibility

Samples from the same specimens were analysed over several batches, with the reproducibility given as the s.d. (±1 s.d.). Statistical significance tests were conducted using either a Welch's $t$-test, given a similar sample size and a heterogeneous variance, or an individual $t$-test for similar sample sizes and variances[123,124]. All analyses were conducted using Python3 on a Jupyter Notebook (v.5.7.4). The data were imported using the Pandas library and plotted using the Seaborn or Matplotlib libraries.

The nitrogen isotope ratios ($\delta^{15}N$) were determined using a purpose-built inlet system coupled to a Thermo MAT253 Plus stable isotope ratio mass spectrometer (running Isodat v.3.0 software). The carbon and oxygen isotope ratios ($\delta^{13}C$ and $\delta^{18}O$, respectively) were measured by an IRMS (Delta V Advantage, Thermo Scientific) connected to a GasBench II unit (Thermo Scientific) (running Isodat v.3.0 software).

## Reporting summary

Further information on research design is available in the Nature Portfolio Reporting Summary linked to this article.

## Data availability

All data are available in Supplementary Tables 2–5 and an Excel datafile will be stored in Pangaea (https://issues.pangaea.de/browse/PDI-38892).

## Code availability

Codes used for the figures and data analyses are available on GitHub (https://github.com/marinejon/Coral-Photosymbiosis-on-Mid-Devonian-Reefs.git).

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

**Acknowledgements** This study was funded by the Max Planck Society. We thank B. Hinnenberg, M. Schmitt and F. Rubach for their technical support during the sample preparation and analyses. We thank E. L. Murphy for proofreading the original manuscript and providing valuable

suggestions. We thank N. Prawitz for preparing the thin sections from the Hagen-Balve Reef samples. We thank M. Ricker for his support in the Palaeozoic coral collection at Senckenberg. We thank Z. S. Aboussalam and R. T. Becker for access to conodont material from Binolen. J.J. acknowledges the support of the Max Planck Graduate Center. A.M.-G. acknowledges Deutsche Forschungsgemeinschaft (German Research Foundation) Project number 468591845–SPP 2299/ Project number 441832482, and N.N.D. acknowledges a Paul Crutzen post-doctoral fellowship. D.M.S. acknowledges the Tuttle Fund of the Department of Geosciences at Princeton University. We extend our gratitude to the Paul Ungerer Foundation for their generous financial support, which enabled a previous fieldtrip to Western Sahara.

**Author contributions** J.J., S.F.Z. and A.M.G. designed the project. J.J. sampled the material, conducted the $\delta^{15}N$ analysis in the laboratory of A.M.G., performed the statistical analyses and wrote the main body of the manuscript, with help from A.M.G. and D.M.S. S.F.Z. performed the fieldwork at Binolen, collected the fossil samples, provided the thin sections and made the taxonomic determinations of the Mid-Devonian corals. T.S. contributed conodont fossils for the CAI and helped with the interpretation of diagenetic overprinting. N.D. and A.F. were involved in the analytical training of J.J. H.V. provided the carbon and oxygen isotope data. S.M. generated the palaeogeographic map. A.M. provided access to the Palaeozoic coral collection and approved the sampling. E.S. provided Mid-Devonian material from the Eifel region, Morocco and Western Sahara and contributed to all matters Devonian. T.W. contributed part of the modern coral samples and the modern nitrate maps. D.J. provided recent coral samples for comparative analysis. A.M.G. supervised the project and the geochemical analyses of the samples. All authors were involved in discussion of the data at different stages of the project and contributed to the final version of the manuscript.

**Funding** Open access funding provided by Max Planck Society.

**Competing interests** The authors declare no competing interests.

**Additional information**
**Correspondence and requests for materials** should be addressed to Jonathan Jung or Alfredo Martínez-García.

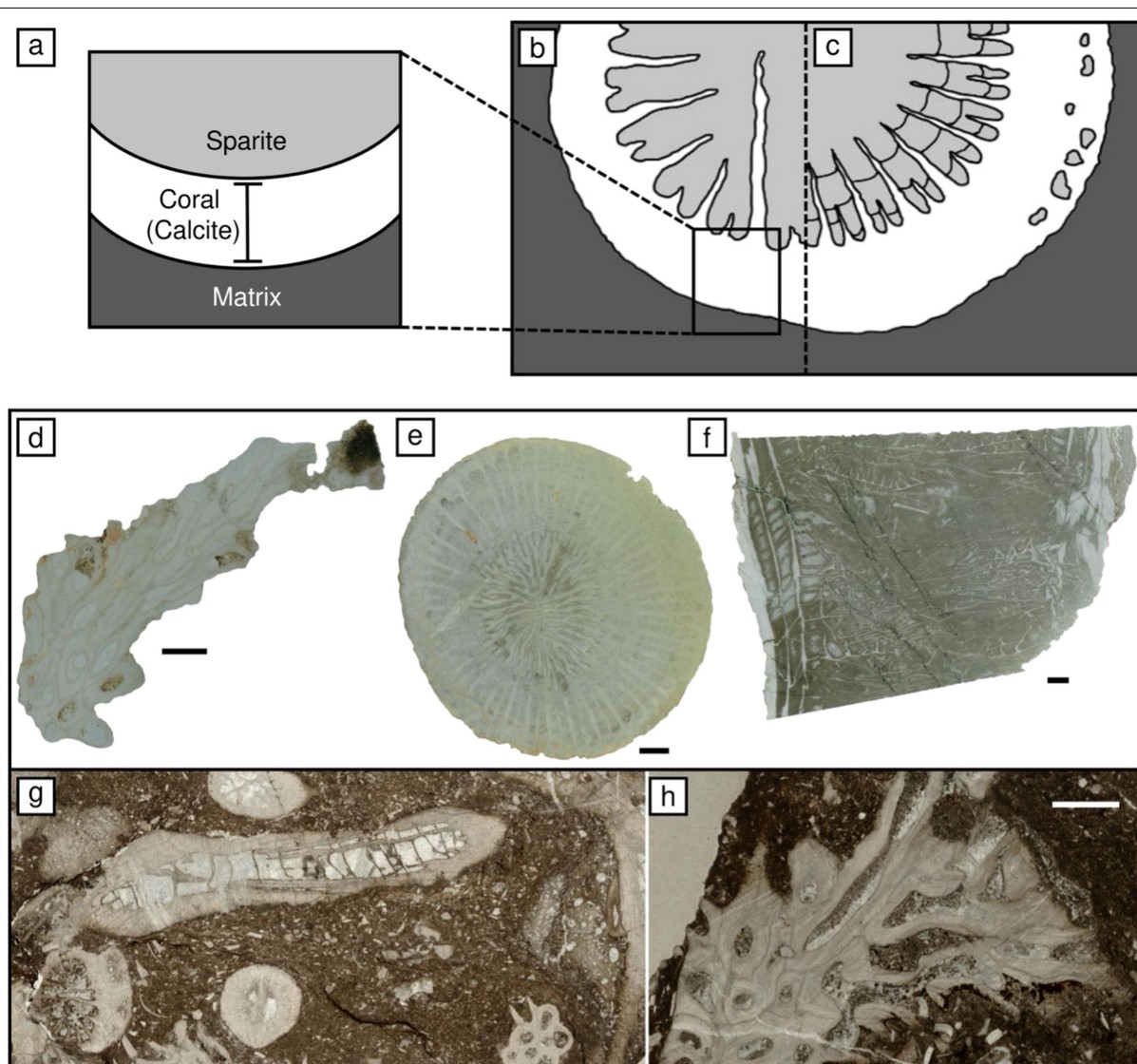

**Extended Data Fig. 1 | Schematic sampling strategy and representative thin sections of Paleozoic corals. a**) Schematic representation of phases (sedimentary matrix, coral skeleton, sparite), which were carefully extracted. Cross section with distribution of phases in rugose corals **b**) schematic of *Dendrostella trigemme* and **c**) *Temnophyllum latum*. Note that each phase may yield contamination from the other, due to the skeletal architecture of analyzed corals (e.g., intra-skeletal sparite content). Microphotographs of thin sections (under transmitted light) of selected Givetian corals from **d**) the Blankenheim Syncline (Eifel Mountains, Germany), **e**)-**f**) Tafilalt (Morocco), and **g**)-**h**) Hagen Balve Reef (Sauerland, Germany). **d**) Tabulate (auloporid) coral *Roemerolites brevis brevis* (SMF40160). **e**) Solitary rugosa coral *Acanthophyllum concavum* (SMF75854). **f**) Solitary rugose coral *Siphonophrentis* sp. (SMF75855). **g**) Longitudinal and cross sections of fasciculate (dendroid) rugosa coral *Dendrostella trigemme* and fragments of the tabulate (auloporid) coral *Roemerolites brevis rhiphaeus* (GMM B2C.59-4). One cross section of *D. trigemme* (lower left) shows species-specific dendroid astogeny. **h**) Anastomosing branch of *R. brevis rhiphaeus* (GMM B2C.59-2), showing partial infilling of corallites by sparite and/or carbonate sediment. All scale bars: 2 mm.

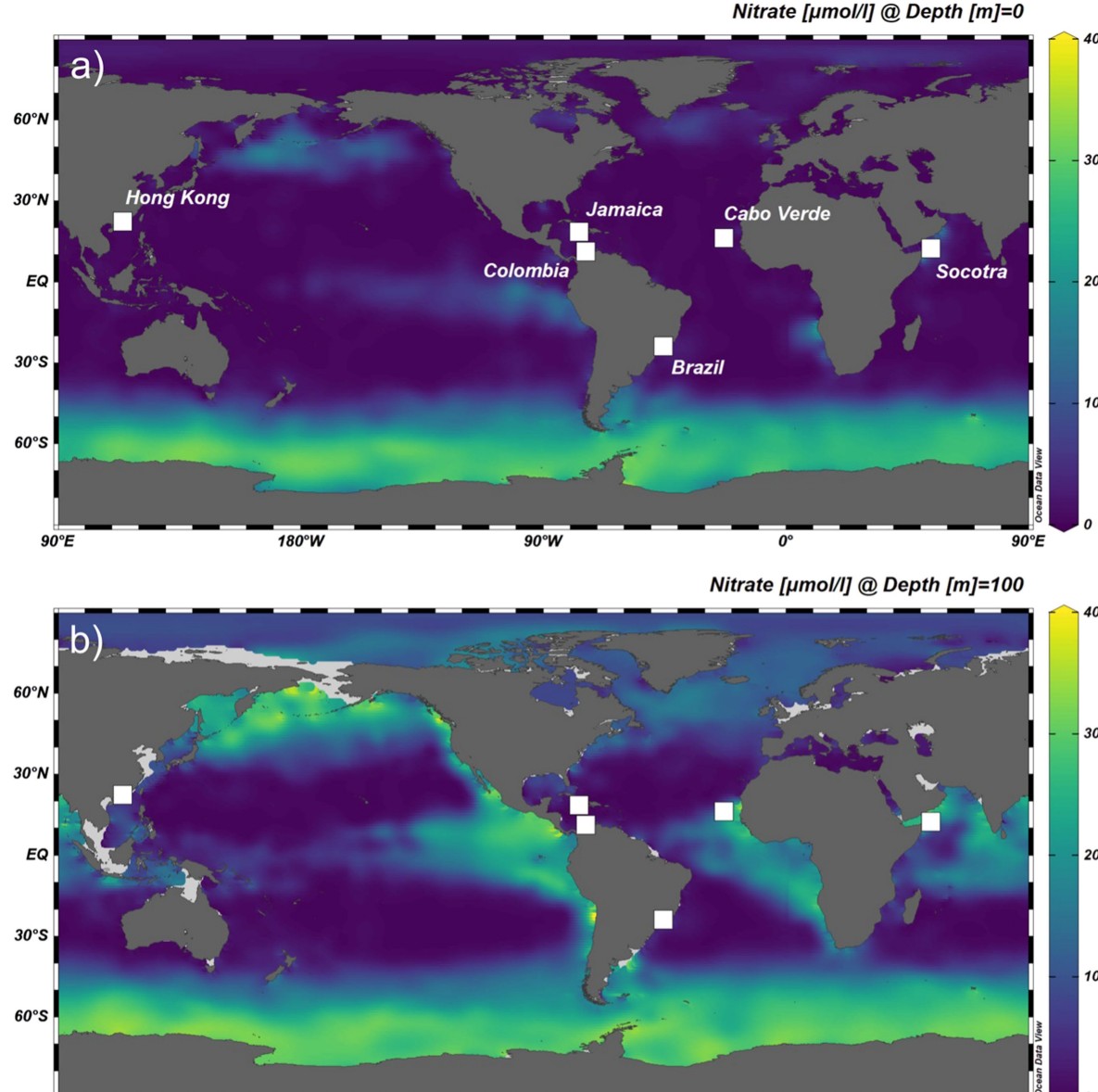

**Extended Data Fig. 2 | Modern reef locations with respect to (sub-)surface nitrate concentrations. a**) Global surface nitrate concentrations with respect to of modern symbiont-bearing and symbiont-barren coral pairs. **b**) Global nitrate concentrations at 100 m depth. Visualizations were created with Ocean Data View Version 5.6.3 (Schlitzer, Reiner, Ocean Data View, https://odv.awi.de, 2023).

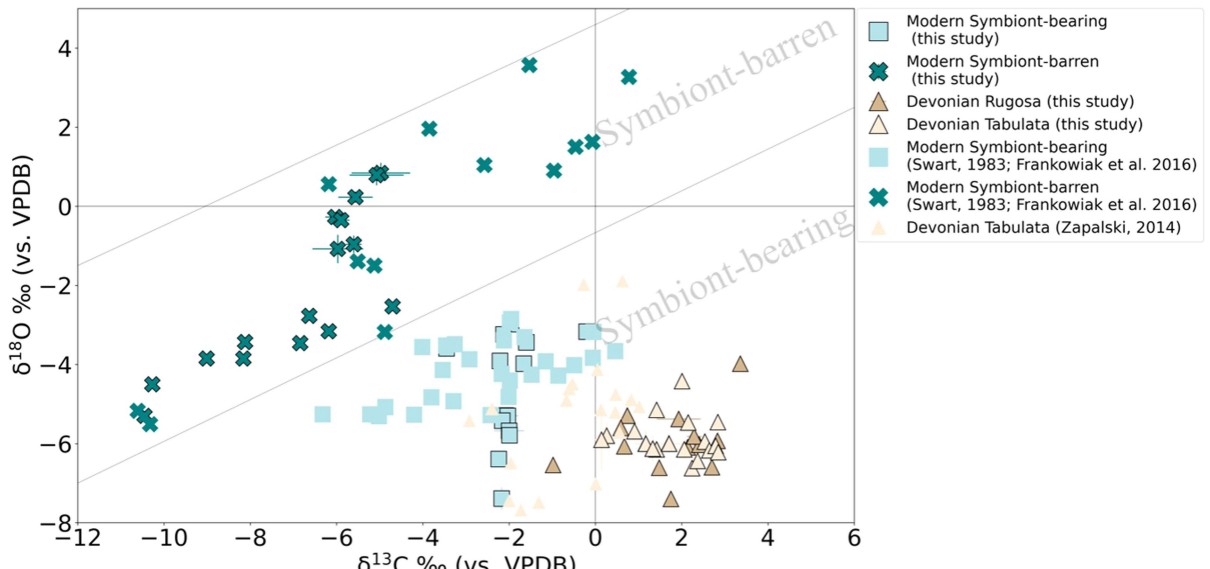

**Extended Data Fig. 3 | Oxygen and carbon isotope cross plot of modern and Paleozoic corals.** Scatterplot of oxygen and carbon isotope values measured on Paleozoic rugose and tabulate corals and their respective sedimentary matrix from the Hagen Balve Reef and Eifel region. Modern analogues of symbiont-bearing (light green squares) and symbiont-barren (dark green crosses) species from various locations were measured. A compilation of previously measured modern and fossil symbiont-bearing and symbiont-barren corals are also included from Swart (1983), and Frankowiak et al.[31] and Zapalski 2014, where samples indicate distinct spaces for symbiont-bearing and symbiont-barren species as indicated by the respective field (adapted from Swart, 1983). Devonian values are shown as beige triangles, with darker and lighter brown tones indicating rugose and tabulate species, respectively. Notably, all Paleozoic samples from this study cluster within the symbiotic range and would indicate no difference between coral groups. Replicate measurements of the same species are taken together and the respective variation (±1 SD) is shown by horizontal (oxygen isotopes) or vertical lines (carbon isotopes).

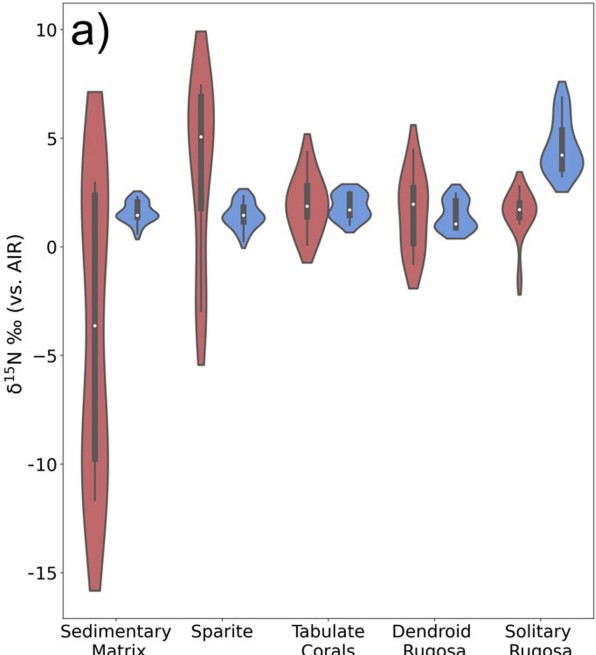 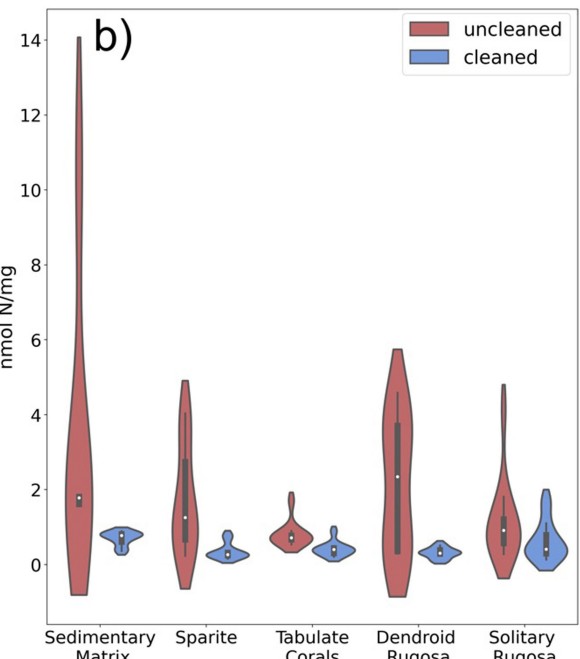

**Extended Data Fig. 4 | Comparison of cleaned and uncleaned material from Sauerland.** Coral-bound nitrogen isotope values of three distinct coral groups and their surrounding material from the Hagen Balve Reef (in ‰ vs. air). Cleaned material (sedimentary matrix: n = 16, sparite: n = 20, tabulate corals: n = 10, dendroid rugose corals: n = 13, solitary rugose corals: n = 15) and uncleaned material (sedimentary matrix: n = 10, sparite: n = 12, tabulate corals: n = 10, dendroid rugose corals: n = 9, solitary rugose corals: n = 13) yield a significant difference for the sediment (F = 837.56, $P$ = 0.03) and solitary rugose corals (F = 0.91, $P$ = 3.6e-07) but not for sparite (F = 17.23, $P$ = 0.09), dendroid rugose corals (F = 7.57, $P$ = 0.83), or tabulate corals (F = 4.76, $P$ = 0.79). **b)**

Corresponding weight-normalized nitrogen content (in nmol N per mg of carbonate powder) is given. Overall, nitrogen content is very low but always higher in unclean samples. Mean values are indicated by the white dots. The lower and upper hinges indicate the first and third quartiles, encapsulating the inter-quartile range (IQR). The whiskers extend to the smallest and largest values within 1.5 times the IQR from the hinges, depicting the spread of the data. The shape of the violinplot is defined by a kernel density estimate (KDE). Statistical significance tests were conducted either by a Welch's t-test given a similar sample size and a heterogenous variance (indicated by F ≥ 1) or an individual t-test for similar sample sizes and variances (indicated by F ≤ 1).

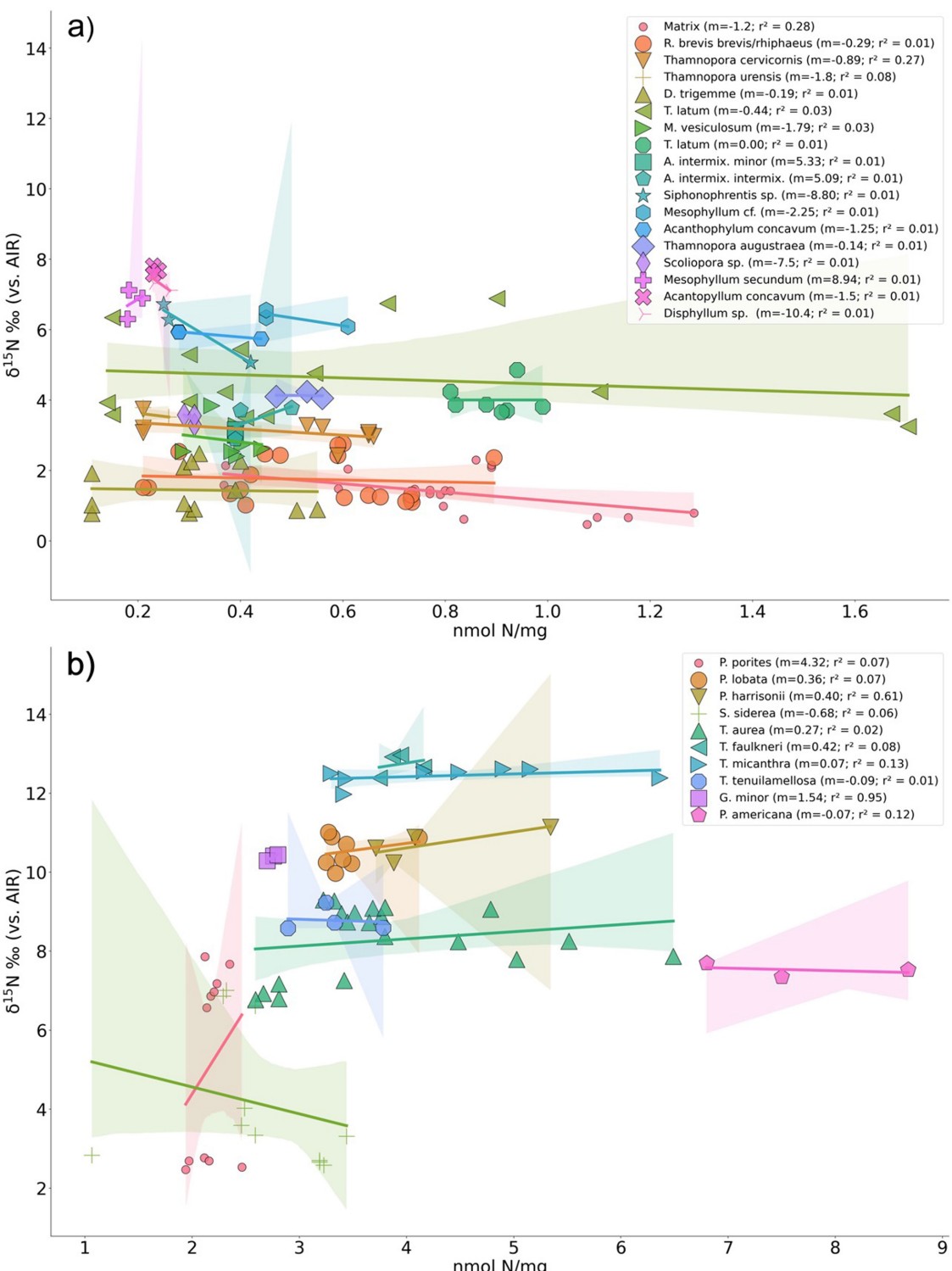

**Extended Data Fig. 5 | Cross plot of nitrogen isotope and nitrogen content values of Paleozoic and modern corals. a**) The relationship between nitrogen content per mg of cleaned carbonate powder and the nitrogen isotope composition of each species that was analyzed in the recent reefs. For each species, the slope (m) and correlation coefficient (r²) are given. Each colored envelope represents 95% confidence interval of the regression. **b**) The relationship between nitrogen content per mg of cleaned carbonate powder and the nitrogen isotope composition of each species that was analyzed on the Mid-Devonian reefs. For each species, the slope (m) and correlation coefficient (r²) are given.

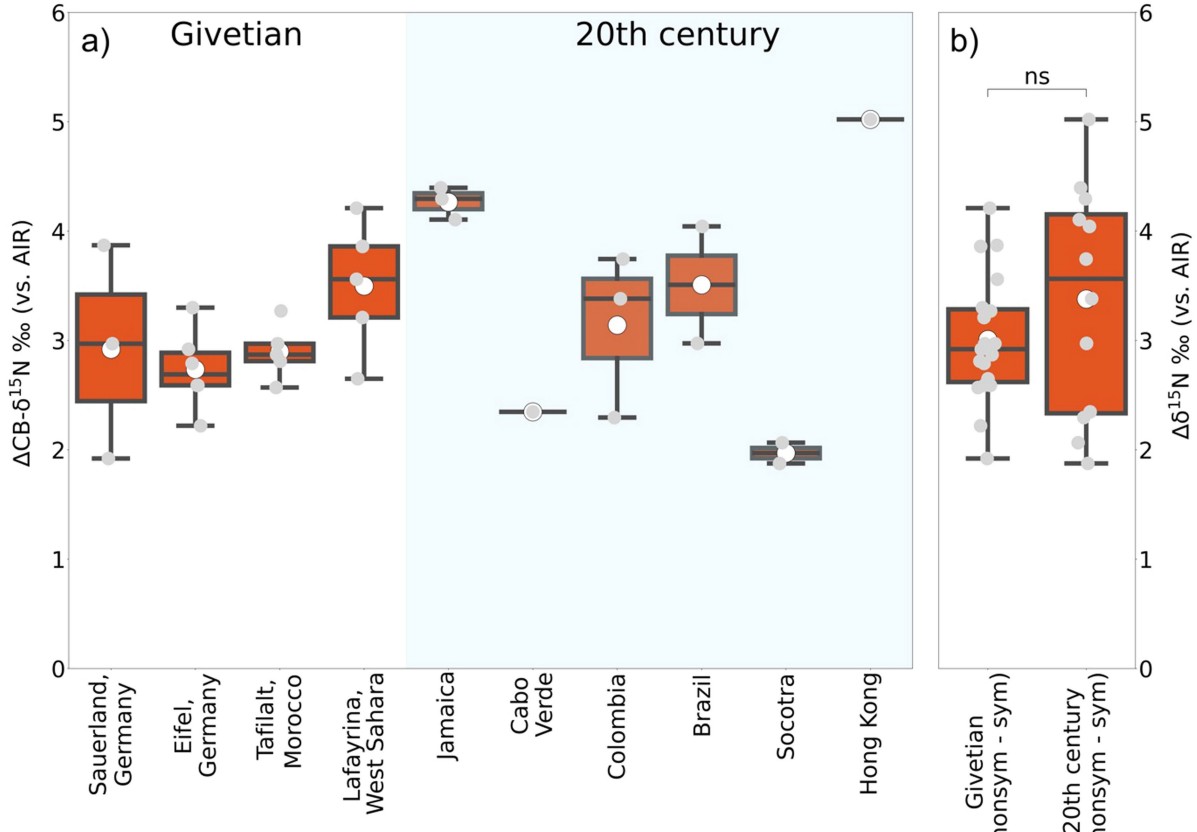

**Extended Data Fig. 6 | Average nitrogen isotope difference from Paleozoic and modern corals at individual locations. a**) Average N isotopic differences between solitary and tabulate/fasciculate rugose species from the Mid-Devonian (Givetian) compared to the difference between modern symbiont-bearing and symbiont-barren corals. The white dot represents the average value whereas the middle line represents the median value. The lower and upper bound of the box correspond to the first and third quartiles. The upper whisker extends from the upper bound box to the largest value within 1.5 times the inter-quartile range (IQR) from the hinge, while the lower whisker extends from the lower bound box to the smallest value within 1.5 times the IQR from the hinge. Values beyond the whiskers are considered outliers and are plotted individually. **b**) Average differences between symbiont-barren and -bearing species based on location and expressed as $\Delta\delta^{15}N = \delta^{15}N_{non-sym.} - \delta^{15}N_{sym}$. Differences vary between 1.97‰ and 5.02‰ for all Givetian and modern samples. A Welch's t-test for unequal variances indicates that there is no significant difference between the values of $\Delta\delta^{15}N$ from ancient reefs and the $\Delta\delta^{15}N$ differences between the co-occurring symbiont-bearing and -barren corals from modern reefs ($F = 0.01$, $P = 0.27$).

# Reporting Summary

Please do not complete any field with "not applicable" or n/a.  Refer to the help text for what text to use if an item is not relevant to your study.
For final submission: please carefully check your responses for accuracy; you will not be able to make changes later.

## Statistics

For all statistical analyses, confirm that the following items are present in the figure legend, table legend, main text, or Methods section.

| n/a | Confirmed | |
|---|---|---|
| ☐ | ☒ | The exact sample size (*n*) for each experimental group/condition, given as a discrete number and unit of measurement |
| ☐ | ☒ | A statement on whether measurements were taken from distinct samples or whether the same sample was measured repeatedly |
| ☐ | ☒ | The statistical test(s) used AND whether they are one- or two-sided<br>*Only common tests should be described solely by name; describe more complex techniques in the Methods section.* |
| ☒ | ☐ | A description of all covariates tested |
| ☒ | ☐ | A description of any assumptions or corrections, such as tests of normality and adjustment for multiple comparisons |
| ☐ | ☒ | A full description of the statistical parameters including central tendency (e.g. means) or other basic estimates (e.g. regression coefficient) AND variation (e.g. standard deviation) or associated estimates of uncertainty (e.g. confidence intervals) |
| ☐ | ☒ | For null hypothesis testing, the test statistic (e.g. *F*, *t*, *r*) with confidence intervals, effect sizes, degrees of freedom and *P* value noted<br>*Give P values as exact values whenever suitable.* |
| ☒ | ☐ | For Bayesian analysis, information on the choice of priors and Markov chain Monte Carlo settings |
| ☒ | ☐ | For hierarchical and complex designs, identification of the appropriate level for tests and full reporting of outcomes |
| ☒ | ☐ | Estimates of effect sizes (e.g. Cohen's *d*, Pearson's *r*), indicating how they were calculated |

*Our web collection on statistics for biologists contains articles on many of the points above.*

## Software and code

Policy information about availability of computer code

| | |
|---|---|
| Data collection | All analyses were conducted using Python3 on a Jupyter Notebook® (version 5.7.4). Data were imported using the Pandas library and plotted with Seaborn or Matplotlib libraries.<br>The nitrogen isotopes were determined by a purpose-built inlet system coupled to a Thermo MAT253 Plus stable isotope ratio mass spectrometer (software: Isodat version 3.0). Oxygen and carbon isotopes were were measured with an isotope ratio mass spectrometer (IRMS) (Delta V Advantage, Thermo Scientific, Bremen, Germany) which is connected to a GasBench II unit (Thermo Scientific) (software: Isodat version 3.0). |
| Data analysis | No software was used for data analysis and the codes used for figures and data analyses are available on GitHub (https://github.com/marinejon/Coral-Photosymbiosis-on-Mid-Devonian-Reefs.git) |

For manuscripts utilizing custom algorithms or software that are central to the research but not yet described in published literature, software must be made available to editors and reviewers. We strongly encourage code deposition in a community repository (e.g. GitHub). See the Nature Portfolio guidelines for submitting code & software for further information.

## Data

Policy information about availability of data

All manuscripts must include a data availability statement. This statement should provide the following information, where applicable:

- Accession codes, unique identifiers, or web links for publicly available datasets
- A description of any restrictions on data availability
- For clinical datasets or third party data, please ensure that the statement adheres to our policy

All data are available in the Supplementary Tables 2 - 5 and upon publication data will be stored in Pangaea (https://www.pangaea.de/)

## Research involving human participants, their data, or biological material

Policy information about studies with human participants or human data. See also policy information about sex, gender (identity/presentation), and sexual orientation and race, ethnicity and racism.

| | |
|---|---|
| Reporting on sex and gender | N/A |
| Reporting on race, ethnicity, or other socially relevant groupings | N/A |
| Population characteristics | N/A |
| Recruitment | N/A |
| Ethics oversight | N/A |

Note that full information on the approval of the study protocol must also be provided in the manuscript.

# Field-specific reporting

Please select the one below that is the best fit for your research. If you are not sure, read the appropriate sections before making your selection.

☐ Life sciences  ☐ Behavioural & social sciences  ☒ Ecological, evolutionary & environmental sciences

For a reference copy of the document with all sections, see nature.com/documents/nr-reporting-summary-flat.pdf

# Ecological, evolutionary & environmental sciences study design

All studies must disclose on these points even when the disclosure is negative.

| | |
|---|---|
| Study description | Pairs of modern symbiont-bearing and symbiont-barren samples were analyzed to define the isotopic offset (carbon, oxygen and nitrogen isotopes) between the two ecological groups. Samples of Paleozoic groups and different morphologies were analyzed for carbon, oxygen and nitrogen isotopes and compared to the modern pairs. Carbon and oxygen isotopes were used because of prior studies and to show that the direct comparison between modern and Paleozoic corals does not yield a comprehensive picture to differentiate between ancient symbiont-bearing and -barren species. Nitrogen isotopes analyses were used as a new way to distinguish between ecological groups of the Paleozoic. |
| Research sample | Fossilized coral samples of the groups Tabulata and Rugosa from the Paleozoic taken from the Sauerland and Eifel region in Germany as well as Tafilalt in Morocco and Sabkhat Lafayrina in West Sahara. Furthermore, we analyzed modern scleractinian corals from several locations as indicated in Extended Data Figure 2 and Supplementary Table 3 & 5. |
| Sampling strategy | For the Paleozoic samples, we used specimens that provided sufficient material to compare uncleaned and cleaned material (as indicated in Figure 2 and Extended Data Figure 4). Sample material was drilled with a hand-dremel (0.9 mm drill-bit) and material was only taken from the calcitic phase (as indicated in Extended Data Figure 1). For modern samples, material was drilled from samples that were taken from the same reef location and depth. Coral pieces (~2x2 cm) were cut and crushed in an Achat mortar before being cleaned. |
| Data collection | Nitrogen Isotope data were analyzed at the Stable Isotope Laboratory of the Max-Planck Institute for Chemistry in Mainz, Germany. Data were mainly analyzed by Jonathan Jung with the supervision of Alfredo-Martinez-Garcia. Carbon and Oxygen Isotopes were analyzed in the inorganic stable isotope laboratory at the Max-Planck-Institute for Chemistry in Mainz, Germany. |
| Timing and spatial scale | Isotopic analysis were conducted between October 2019 and December 2023. Paleozoic coral samples were taken from two locations in Germany with different diagenetic histories as well as two locations from the opposite basin of the Rheic Ocean (Morocco and West Sahara) and compared to modern coral samples from a range of underlying environmental conditions from which we expected different nitrogen isotope values for symbiont-bearing corals. Our rational was to show that despite different |

underlying environmental conditions (i.e., oligotrophic vs. eutrophic) the difference between symbiont-bearing and symbiont-barren nitrogen isotope values stays the same.

| | |
|---|---|
| Data exclusions | No samples were excluded. |
| Reproducibility | Samples of the same specimens were analyzed over several batches and reproducibility is given as the standard deviation. |
| Randomization | Samples did not need to be grouped since there are no inter-dependencies between species. |
| Blinding | Samples were analyzed in a random order but had to be indicated by labels. |

Did the study involve field work?  ☒ Yes  ☐ No

## Field work, collection and transport

| | |
|---|---|
| Field conditions | At Sauerland, hand pieces were collected off the ground. Collection happened in October 2018, while It was cloudy but dry at 13°C. |
| Location | The main material was collected near a cliff at the top of the Binolen section ("C-Layers" after Löw et al., 2022; GPS: 51°22'12''N, 7°51'27''E) within the Hönne Valley in north-western Sauerland. |
| Access & import/export | Samples were taken close to a publicly accessible forest trail where no permits are needed. Pieces were exclusively collected from the ground. |
| Disturbance | Only those hand-pieces were collected that could be picked up from the ground. |

# Reporting for specific materials, systems and methods

We require information from authors about some types of materials, experimental systems and methods used in many studies. Here, indicate whether each material, system or method listed is relevant to your study. If you are not sure if a list item applies to your research, read the appropriate section before selecting a response.

## Materials & experimental systems

| n/a | Involved in the study |
|---|---|
| ☒ | ☐ Antibodies |
| ☒ | ☐ Eukaryotic cell lines |
| ☐ | ☒ Palaeontology and archaeology |
| ☒ | ☐ Animals and other organisms |
| ☒ | ☐ Clinical data |
| ☒ | ☐ Dual use research of concern |
| ☒ | ☐ Plants |

## Methods

| n/a | Involved in the study |
|---|---|
| ☒ | ☐ ChIP-seq |
| ☒ | ☐ Flow cytometry |
| ☒ | ☐ MRI-based neuroimaging |

## Palaeontology and Archaeology

| | |
|---|---|
| Specimen provenance | Sampling permits for specimens from the the Senckenberg Research Institute and Natural History Museum Frankfurt, Germany were issued. No permits were needed otherwise. |
| Specimen deposition | Thin sections for species identification from the initial Hagen Balve Reef at Binolen will be stored in the Geomuseum of the Westfälische Wilhelms University in Münster (GMM) under the inventory numbers GMM B2C.59-1 to GMM B2C.59-9. Sample material from the Eifel and the modern samples are stored at the Senckenberg Research Institute and Natural History Museum Frankfurt, Germany. All sample aliquots are stored at the Max-Planck Institute for Chemistry in Mainz. |
| Dating methods | No new dates were obtained. |

☐ Tick this box to confirm that the raw and calibrated dates are available in the paper or in Supplementary Information.

| | |
|---|---|
| Ethics oversight | No ethical approval was necessary given that no endangered species were analyzed. |

Note that full information on the approval of the study protocol must also be provided in the manuscript.

# Plants

| | |
|---|---|
| Seed stocks | *Report on the source of all seed stocks or other plant material used. If applicable, state the seed stock centre and catalogue number. If plant specimens were collected from the field, describe the collection location, date and sampling procedures.* |
| Novel plant genotypes | *Describe the methods by which all novel plant genotypes were produced. This includes those generated by transgenic approaches, gene editing, chemical/radiation-based mutagenesis and hybridization. For transgenic lines, describe the transformation method, the number of independent lines analyzed and the generation upon which experiments were performed. For gene-edited lines, describe the editor used, the endogenous sequence targeted for editing, the targeting guide RNA sequence (if applicable) and how the editor was applied.* |
| Authentication | *Describe any authentication procedures for each seed stock used or novel genotype generated. Describe any experiments used to assess the effect of a mutation and, where applicable, how potential secondary effects (e.g. second site T-DNA insertions, mosiacism, off-target gene editing) were examined.* |

