## [Peer Review File · Nature]

Manuscript Title: Coral Photosymbiosis on Mid-Devonian Reefs

Editorial Notes:

Redactions – Third Party Material

Reviewer Comments & Author Rebuttals

Reviewer Reports on the Initial Version:

Referees' comments:

Referee #1 (Remarks to the Author):

The paper by Jung et al. "Coral Photosymbiosis on Mid-Devonian Reefs" presents results on the studies of nitrogen stable isotopes of Palaeozoic coral skeletons (representatives of Rugosa and Tabulata, comparatively with Scleractinian) in relation to photosymbiosis. The paper is very interesting and in fact represents a long-awaited study.

This is the first study of the nitrogen isotopes in Palaeozoic corals aiming at resolving of a long-standing enigma of whether Palaeozoic corals possessed photosymbionts. Other lines of evidence (morphology, growth patterns and C and O stable isotopes) have already been used in photosymbiosis assessment in previous studies, but the study of nitrogen isotopes adds a new dimension to the discussion. However, contrary to what the authors state, however (line 22) it is not the oldest isotopic constraint on the evolution of photosymbiosis. A study by Zapalski 2014 on the carbon and oxygen stable isotopes demonstrated possible photosymbiosis as early as in the Silurian. Nonetheless the study is of a very broad importance, as it adds probably the most important line of evidence on algal symbionts of Tabulata and Rugosa.

To my knowledge (I had only a little experience with geochemistry) the methodologies of processing samples and isotope analyses are correct. Statistics are appropriately used. I feel, however, that the paper does not explain (or justify) why these particular taxa of Devonian corals were chosen, and the most important ecological and taxonomical groups of corals contributing to Devonian reefs are not represented in this study.

Rugose corals were subordinate contributors to mid-Palaeozoic reefs; on the contrary, at least three orders of tabulate corals formed a plethora of morphologies, comparable to those of scleractinians, with diversified ecological characteristics. Authors analyze five taxa of tabulate corals and only one tabulate. What is more, Roemerolites, the only analysed tabulate coral, was an unusual and rare organism in Devonian reef ecosystems. Auloporids rarely formed branching colonies, while the most prevalent Devonian branching corals were pachyporids (e.g. Lecompte 1939, Hubert et al. 2007). Current sampling pattern is that the study analyses five taxa of accessory reef faunas and only a single taxon of the reef-forming corals

My feeling is that the manuscript suffers from lack of Palaeozoic coral expertise. I would strongly suggest extending the analysis to cover most common reef-building and reef-dwelling tabulates (with emphasis

on massive, platy and branching morphotypes), especially that platy (mesophotic) morphotypes were undoubtedly photosymbiotic, as such a morphology does not occur in azooxanthellates (e.g. Zapalski & Berkowski 2019). This would give additional control on the isotope analysis results. Additionally, some reef-dwelling rugose corals (massive, cerioid) could be added to the analysis. With coverage of at least 10-12 taxa the results will be sound enough to claim the solution of the problem stated in the manuscript title.

The analysis contains a number of scleractinians used for comparative study. In other words the sampling in this work is unbalanced: over a dozen of scleractinian taxa and conodonts were sampled, but they form only auxiliary data (regardless of the importance of these data). There are five rugose coral taxa, but these were accessory elements of Devonian reefs. The truly reef-forming forms are represented by a single, not very typical taxon. I may conclude that in the present version of the manuscript the data are interesting, but to large extent incomplete.

At present, the title implies reference to a large portion of Devonian reef corals, but this is rather a case study. In my opinion the paper has great potential and the tools used in this work may bring answers to key questions about the paleoecology of Palaeozoic reefs, but to do so the study should cover more groups of corals. This should be easily done, as potential material for studies is available in numerous institutions in Belgium, Germany, France, Poland or Sweden.

As the paper is very interesting and after amendments it can make a significant contribution to palaeontology, marine biology and environmental sciences I recommend accepting it after a major revision, given the additional analyses should be done.

As the paper needs quite important changes, I have selected only several minor comments:

Line 42 – please add Burchette 1981 – major review work on Devonian reefs of Europe.

Line 73-75 – Results of molecular data published by McFadden et al (2021) suggest the appearance of photosymbiosis as early as Devonian. This is postdates somewhat data from morphology, growth patterns and C & O stable isotopes (Zapalski 2014, Zapalski & Berkowski 2019).

Line 79 – authors state that solitary corals are rarely photosymbiotic. While partly true, this statement should be developed and nuanced, as there is a number of solitary representatives of Fungiidae and Mussidae that are zooxanthellate. Certainly when it comes to a number of taxa the azooxanthellates prevail, but on the other hand they are very rare in reefal environments.

Line 80-88 – This section suggests that majority of works suggest absence of photosymbionts in Palaeozoic corals, which is not correct. Evidence in favour of photosymbiosis drawn from morphology and growth patterns was presented by Coates and Jackson (1987), Zapalski et al. 2017, Zapalski and

Berkowski 2019. While the stable isotope results by Zapalski (2014) were questioned by Jakubowicz et al. (2015) the latter authors based their study on rugose corals that had high-Mg calcite skeletons, much more prone to diagenesis than low-Mg skeletons of tabulates, studied by Zapalski (2014). The question is therefore whether the study by Jakubowicz et al. (2015) is compatible with that of Zapalski (2014). What is more, the results of the auxiliary oxygen and carbon isotope sampling done by the authors (Supplementary Table 4) are very similar to those published by Zapalski (2014).

Line 179-183. Again, authors state that “This is thus the oldest geochemical expression of the presence of photosymbiotic associations in corals to date” which statement is imprecise, as it is concordant with the results from oxygen and carbon isotopes (Zapalski 2014). Authors also state that their study is expanding the record of photosymbiosis by 170 Ma, but evidence for photosymbiosis is known from the Silurian (430 ma, Zapalski 2014, Zapalski & Berkowski 2019).

Lines 446-449 – the phrase “the narrow range of $\delta^{18}\text{O}$ and $\delta^{13}\text{C}$ values would suggest photosymbiosis across species^{24,44,112} and thus, would stand in direct contrast to our result from CB-447 $\delta^{15}\text{N}$ measurements, and previous suggestions, based on morphological analysis, that no Paleozoic corals harbored symbionts^{4,5}” is somewhat unclear – the results obtained by the authors are generally concordant with the previously published data, or I missed something here.

BIBLIOGRAPHY (I list here only papers not cited in the original contribution)

Burchette, T. P. European Devonian reefs: a review of current concepts and models. *SEPM Spec. Public.* 30, 85–142 (1981).

Hubert, B. L., Zapalski, M., Nicollin, J. P., Mistiaen, B., & Brice, D. (2007). Selected benthic faunas from the Devonian of the Ardennes: an estimation of palaeobiodiversity. *Acta Geologica Polonica*, 57(2), 223-262.

McFadden CS, Quattrini AM, Brugler MR, Cowman PF et al. Phylogenomics, Origin, and Diversification of Anthozoans (Phylum Cnidaria). *Syst Biol.* 2021 Jun 16; 70(4):635-647 (2021).

Lecompte M. J. (1939) Les tabulés du Dévonien moyen et supérieur du bord sud du bassin de Dinant. *Musée Royal D’histoire Naturelle De Belgique* 90, 1–229.

Zapalski, M.K., Wrzołek, T., Skompski, S. et al. Deep in shadows, deep in time: the oldest mesophotic coral ecosystems from the Devonian of the Holy Cross Mountains (Poland). *Coral Reefs* 36, 847–860 (2017).

Zapalski, M.K., Berkowski, B. The Silurian mesophotic coral ecosystems: 430 million years of photosymbiosis. *Coral Reefs* 38, 137–147 (2019).

Mikolaj K. Zapalski
University of Warsaw

Referee #2 (Remarks to the Author):

This is a remarkable study that substantially increases our understanding of the origins of some of the most important symbiotic associations on Earth - those that underpin the immense productivity and success of coral reefs in deep time. The authors are amongst the best in the world at making painstaking measurements of the $\delta^{15}\text{N}$ of skeletal bound organic matter from fossil and modern coral specimens - through a clever method that has been pioneered and refined by several of the authors over the years. When those first methodological papers emerged it was obvious that the huge potential was to unlock an entirely new field of paleobiological reconstructions - not only for understanding past ocean processes in terms of chemical and physical paleoceanography - but now also understanding paleoecology.

It is critically important for science to piece together the origins of symbiosis - as we now understand how pervasive biological partnerships are in nature - even within our own human gut. For coral reefs, the symbiosis between dinoflagellate algae and their cnidarian hosts is the “poster-child” for mutualism and one that has a perilous future amidst the Anthropocene. Here, an elegant and meticulous methodology paired with extensive sampling of paleontological and modern collections creates a simple and compelling story that provides insights to where coral reefs come from, and helps us to use that historical context to see more clearly where they are going. The manuscript is of interest to a wide readership - including those interested in symbiosis, and bio geosciences.

As an isotopist myself I am best able to evaluate the $\delta^{15}\text{N}$ data component of this study and I find the results to be compelling. The manuscript clearly illustrates how the issue of organic contamination is mitigated. The remarkable consistency of pattern with low $\delta^{15}\text{N}$ in modern photosymbiotic corals similar to ancient colonial taxa is impressive. There is a robust signal - again consistent across deep time - seen in exclusively heterotrophic modern corals and their ancient counterparts. That the difference between symbiotic and aposymbiotic species hovers around 3‰ is also quite profound as this is around the generalization that around 3.4‰ is the average difference between consumers and their diet.

This paper thus has several key advances/outcomes that go well beyond refining the temporal origins of symbiosis in the sea.

Namely,

1. it confirms the high fidelity of biological recorders - in contrast to sedimentary records from mixed (abiotic and biotic) origins.
2. It firmly validates $\delta^{15}\text{N}$ as a strong indicator of photosymbiosis and the timing of its emergence as concomitant to widespread oceanic oligotrophy. The former is congruent with many studies of modern corals and the host-symbiont-skeletal compartment and the latter further helps us understand why mitigating local stressors like nutrient pollution is essential for having symbiotic corals.
3. It further establishes that other proxies such as oxygen and carbon isotopes are problematic with confounding diagenetic and kinetic isotope effects.

4. It paves the way for expanding the work through myriad applications in paleo research, including paleo-nutrient reconstructions on carbonate bound deposits on Earth.
5. I am not sure if this is a correct or impactful interpretation - but I found it interesting that the $\delta^{15}\text{N}$ of the reef matrix is more similar to that of the colonial taxa, suggesting that their bio calcification is the major driver of these deposits, as opposed to heterotrophic species.

The manuscript is very well written and clear. Most of the content is accessible to the general readership of the journal and the references therein are suitable for directing the readership to source materials where necessary. I have very few and very minor comments that do not detract from the paper at all, but could help improve it slightly.

1. Symbiodiniaceae is misspelled. Line 55
2. Line 82 - best to avoid using zooxanthellae - an antiquated and outdated term.
3. The manuscript is largely written in the present tense - the past tense should be used particularly when presenting results that were previously measured.
4. Lines 127-145 it would be informative to have sample sizes listed here with the means. It would also be good to understand why the variation for some species/locations is many times greater than other species/locations and whether or not that is an artifact of unbalanced replication.
5. Line 180 - as there is no direct evidence of the presence of symbionts by any corroborative means - the language here should be more deductive. That the patterns for modern symbiotic and aposymbiotic species are similar the authors deduce that the mechanism for this is the presence of photosymbionts that vector DIN to the holobiont.

I whole-heartedly support publication.

David M. Baker

Referee #3 (Remarks to the Author):

I think this is an excellent study of the stable isotope paleoecology of a Devonian ecosystem but I don't think the Devonian data, as presented, paint a clear picture of photosymbiosis in corals. Accordingly I don't think this work in its current form rises to the level of publication in Nature.

The offset between rugose and tabulate corals is indistinguishable from a trophic effect with an offset of ~ 1 trophic level. In this case, the simplest argument is that the average difference of 2.86‰ is simply the result of a trophic isotope effect. This is an interesting paleoecological result but is not particularly strong evidence of photosymbiosis. Rugose corals were large animals and likely had prey that included larger organisms like zooplankton and maybe even small fish that were probably secondary consumers. In this case, it is likely that Rugose corals would have had $\delta^{15}\text{N}$ values that were higher than primary consumers. It is not that I think that tabulates didn't have photosymbionts, it's that I don't think your

isotope data provide an unequivocal case for photosymbiosis in this environment.

You need to include ^{15}N values from the encasing sedimentary strata as opposed to the matrix. I agree that the matrix may be derived from crushed coral materials but you need the context of water column organic matter $\delta^{15}\text{N}$ values. There are many Devonian studies that rely on bulk ^{15}N and many other deep time sections from shallow sequences on continental crust and the general consensus is that the bulk values are probably pretty good reflections of primary processes (Robinson 2012 covers this issue nicely). Like black shales and sapropels (e.g. Mediterranean) the bulk and compound specific values converge on ^{15}N depletion in these organic matter rich deposits and that the values are probably primary. The same logic applies for the Devonian. Sure, bulk values need to be viewed critically, but the body of work out there suggests that they are probably pretty good approximations of the $\delta^{15}\text{N}$ of primary producers. And if we assume that the organic matter in the sediment is composed primarily of phytoplankton (a reasonable assumption) then the tabulates are probably just primary consumers and the rugose are secondary consumers. In any case you really need to report the bulk values and hash out why you don't think they are any good or not.

What is the source of the N in the spar and why does it have the same ^{15}N of the corals?

I know that maybe abstracts don't count as the "first" of some kind of analysis, but there is a AGU abstract from 2017 (Hickey) on Devonian coral nitrogen that comes up when you search on "coral nitrogen isotopes Devonian" in google scholar. It has not been published in any other form as far as I can tell. I also don't think that being first to do something is very interesting and you should probably remove reference to it? Lots of firsts are not better than some piece of science that came afterwards. If the results are in a highly cited journal like Nature, it seems sort of self-evident that it is new.

The conodont temperatures at Hagen on the low end are typical for rocks of this age, but on the high end, that is quite high. In both cases we are talking about rocks that are well past the oil window. Is there any concern that the organic matter in the corals and their pore spaces is derived from migrating oil?

The methods are very detailed and the lengths to which the authors have gone to work with these difficult, nitrogen lean materials is impressive.

Some questions regarding methodological details:

It seems you have done a good job cleaning your samples and the methods are well thought out to be sure you have clean coral carbonate, very nice indeed. Though I am curious about the interiors of the corals. Rugose corals are quite porous and well-preserved specimens can retain the original voids and pores. Are these all filled with spar?

Does the rather higher thermal maturity make the persulfate oxidation difficult? Are you sure that the kerogen or other organic phases are fully oxidized? Can you determine if the nitrogen and carbon

content of the cleaned skeleton match up to the yielded N content in the dissolved sample?

Why do you think that the unclean sedimentary matrix has such incredibly broad $\delta^{15}\text{N}$ values? I have never seen anything like this. A few permil perhaps, but the range is startling.

Were you specifically worried about clay in your drilled samples? I am curious as to why you were concerned about the clay if you are milling out what was hopefully primary carbonate.

Why are you concerned about oxides?

Did you measure the $\delta^{15}\text{N}$ of the sparite or any secondary cements? This may be useful to determine if secondary calcification had any nitrogen from outside sources.

Author Rebuttals to Initial Comments:

Response to Reviews

Referee #1 (Remarks to the Author):

The paper by Jung et al. “Coral Photosymbiosis on Mid-Devonian Reefs” presents results on the studies of nitrogen stable isotopes of Palaeozoic coral skeletons (representatives of Rugosa and Tabulata, comparatively with Scleractinian) in relation to photosymbiosis. The paper is very interesting and in fact represents a long-awaited study.

This is the first study of the nitrogen isotopes in Palaeozoic corals aiming at resolving of a long-standing enigma of whether Palaeozoic corals possessed photosymbionts. Other lines of evidence (morphology, growth patterns and C and O stable isotopes) have already been used in photosymbiosis assessment in previous studies, but the study of nitrogen isotopes adds a new dimension to the discussion. However, contrary to what the authors state, however (line 22) it is not the oldest isotopic constraint on the evolution of photosymbiosis. A study by Zapalski 2014 on the carbon and oxygen stable isotopes demonstrated possible photosymbiosis as early as in the Silurian. Nonetheless the study is of a very broad importance, as it adds probably the most important line of evidence on algal symbionts of Tabulata and Rugosa.

To my knowledge (I had only a little experience with geochemistry) the methodologies of processing samples and isotope analyses are correct. Statistics are appropriately used. I feel, however, that the paper does not explain (or justify) why these particular taxa of Devonian corals were chosen, and the most important ecological and taxonomical groups of corals contributing to Devonian reefs are not represented in this study.

Rugose corals were subordinate contributors to mid-Palaeozoic reefs; on the contrary, at least three orders of tabulate corals formed a plethora of morphologies, comparable to those of scleractinians, with diversified ecological characteristics. Authors analyze five taxa of tabulate corals and only one tabulate. What is more, Roemerolites, the only analysed tabulate coral, was an unusual and rare organism in Devonian reef ecosystems. Auloporids rarely formed branching colonies, while the most prevalent Devonian branching corals were pachyporids (e.g. Lecompte 1939, Hubert et al. 2007). Current sampling pattern is that the study analyses five taxa of accessory reef faunas and only a single taxon of the reef-forming corals

My feeling is that the manuscript suffers from lack of Palaeozoic coral expertise. I would strongly suggest extending the analysis to cover most common reef-building and reef-dwelling tabulates (with emphasis on massive, platy and branching morphotypes), especially that platy (mesophotic) morphotypes were undoubtedly photosymbiotic, as such a morphology does not occur in azooxanthellates (e.g. Zapalski & Berkowski 2019). This would give additional control on the isotope analysis results. Additionally, some reef-

dwelling rugose corals (massive, cerioid) could be added to the analysis. With coverage of at least 10-12 taxa the results will be sound enough to claim the solution of the problem stated in the manuscript title.

The analysis contains a number of scleractinians used for comparative study. In other words the sampling in this work is unbalanced: over a dozen of scleractinian taxa and conodonts were sampled, but they form only auxiliary data (regardless of the importance of these data). There are five rugose coral taxa, but these were accessory elements of Devonian reefs. The truly reef-forming forms are represented by a single, not very typical taxon. I may conclude that in the present version of the manuscript the data are interesting, but to large extent incomplete.

At present, the title implies reference to a large portion of Devonian reef corals, but this is rather a case study. In my opinion the paper has great potential and the tools used in this work may bring answers to key questions about the paleoecology of Palaeozoic reefs, but to do so the study should cover more groups of corals. This should be easily done, as potential material for studies is available in numerous institutions in Belgium, Germany, France, Poland or Sweden.

As the paper is very interesting and after amendments it can make a significant contribution to palaeontology, marine biology and environmental sciences I recommend accepting it after a major revision, given the additional analyses should be done.

We are grateful for the insightful and encouraging comments of R1, which we have addressed by adding samples from three new basins, two of which are from the opposite side of the Rheic Ocean (Tafilalt, Morocco and Sabkhat Lafayrina, Western Sahara). We include six new species of tabulate corals, which come from the two families Pachiporidae and Alveolitidae that the R1 requested. We also expanded the number of rugose corals at each respective site including ceroid species (*Argutastraea* and *Disphyllum*). These species are loosely colonial such that were more completely covering the morphological spectrum from fully colonial to fully solitary corals. While R1 points out that the fully rugose corals were not significant contributors to Devonian reef construction, our goal in measuring these corals was to provide a counterpoint to the corals with colonial or semi-colonial form. As such, our sampling strategy was to have coupled analyses of both forms in any given reef environment. This constrained the number of species and specimens analyzed. Although the method is very sensitive, it is destructive and still requires ~15mg of sample material per analysis. Therefore, we included only those specimens where enough samples were available. In any case, the substantial addition to the study dataset confirms our previous results and strengthens the conclusions of the manuscript.

As the paper needs quite important changes, I have selected only several minor comments:

Line 42 – please add Burchette 1981 – major review work on Devonian reefs of Europe.

We added the citation to the manuscript.

Line 73-75 – Results of molecular data published by McFadden et al (2021) suggest the appearance of

photosymbiosis as early as Devonian. This is postdates somewhat data from morphology, growth patterns and C & O stable isotopes (Zapalski 2014, Zapalski & Berkowski 2019).

We added a sentence about the emergence of photosymbiotic associations in Anthozoa based on phylogenetic reconstructions published by McFadden et al. 2021 and added the respective reference.

Line 79 – authors state that solitary corals are rarely photosymbiotic. While partly true, this statement should be developed and nuanced, as there is a number of solitary representatives of Fungiidae and Mussidae that are zooxanthellate. Certainly when it comes to a number of taxa the azooxanthellates prevail, but on the other hand they are very rare in reefal environments.

We have re-written the statement and use more nuanced language. We have included modern morphological exceptions which suggest that morphology cannot be used as a definitive indicator to distinguish between symbiont-bearing and symbiont-barren coral species.

Line 80-88 – This section suggests that majority of works suggest absence of photosymbionts in Palaeozoic corals, which is not correct. Evidence in favour of photosymbiosis drawn from morphology and growth patterns was presented by Coates and Jackson (1987), Zapalski et al. 2017, Zapalski and Berkowski 2019. While the stable isotope results by Zapalski (2014) were questioned by Jakubowicz et al. (2015) the latter authors based their study on rugose corals that had high-Mg calcite skeletons, much more prone to diagenesis than low-Mg skeletons of tabulates, studied by Zapalski (2014). The question is therefore whether the study by Jakubowicz et al. (2015) is compatible with that of Zapalski (2014). What is more, the results of the auxiliary oxygen and carbon isotope sampling done by the authors (Supplementary Table 4) are very similar to those published by Zapalski (2014).

We addressed this comment by using more nuanced statements and show examples of different interpretations on the association between morphology and photosymbiosis across studies. We measured carbon and oxygen isotopes on every specimen and discuss that in the main text. While we can reproduce results from Zapalski 2014 for tabulate corals, adding rugose samples to the C-O plot yields no differences. This would have suggested that rugose corals also hosted symbionts. As R1 suggests, this may indicate that rugose corals are more prone to diagenesis than tabulate corals, but may also question the comparability of carbon and oxygen isotopes between calcitic Devonian corals and modern aragonitic corals. In any case, we think it is fair to summarize the state of the field as considering carbonate-carbon and -oxygen isotope measurements as ambiguous tool for symbiosis reconstruction. Indeed, this situation motivated the current study.

Line 179-183. Again, authors state that “This is thus the oldest geochemical expression of the presence of photosymbiotic associations in corals to date” which statement is imprecise, as it

is concordant with the results from oxygen and carbon isotopes (Zapalski 2014). Authors also state that their study is expanding the record of photosymbiosis by 170 Ma, but evidence for photosymbiosis is known from the Silurian (430 ma, Zapalski 2014, Zapalski & Berkowski 2019).

We re-wrote the statement to ‘... , which represent a novel isotopic constraint on the evolution of coral photosymbiosis’. In addition, we have removed the sentence about expanding the record of photosymbiosis by 170 Ma. However, in the context of this response, we need to be clear that we disagree with R1’s perspective. We think the literature (as well as our measurements in the supplement; Comms Fig. 1) indicates that neither coral morphology nor carbon or oxygen isotopes are definitive indicators of fossil coral symbiotic status (Coates and Jackson, 1987; Dworzak et al., 2022; Jakubowicz et al., 2015). With regard to isotopic measurements, carbon and oxygen isotope analyses of colonial corals by Zapalski 2014 concluded the widespread existence of photosymbiosis. However, we measured the carbon and oxygen isotopes of other morphologies, including loosely colonial and solitary rugose corals, and find the same ‘symbiotic’ pattern despite the nitrogen isotope measurements arguing against symbiosis and the expectation of R1 that these species are asymbiotic.

Lines 446-449 – the phrase “the narrow range of $\delta^{18}\text{O}$ and $\delta^{13}\text{C}$ values would suggest photosymbiosis across species 24, 44, 112 and thus, would stand in direct contrast to our result from CB-447 $\delta^{15}\text{N}$ measurements, and previous suggestions, based on morphological analysis, that no Paleozoic corals harbored symbionts 4, 5 “ is somewhat unclear – the results obtained by the authors are generally concordant with the previously published data, or I missed something here.

We agree with R1 that this sentence was formulated in an unclear way; we re-wrote it. The observation is that that carbon and oxygen isotope data plot in the same area for both coral groups (Tabulata and Rugosa) on the $\delta^{13}\text{C}$ - $\delta^{18}\text{O}$ plot. The implication of these data (and indeed in all the published data, Comms. Fig. 1) is that, in Paleozoic corals, carbonate carbon and oxygen isotope measurements would suggest photosymbiosis in all corals measured, despite our N isotope evidence for photosymbiosis in the colonial and loosely colonial forms, not in the solitary corals. However, the goal of this paper is not to relitigate the earlier C-O isotope literature. Thus, we address C-O isotope data only in one sentence in the main text.

Comments Figure 1. Scatterplot of oxygen and carbon isotope values measured on Paleozoic rugose and tabulate corals and their respective sedimentary matrix from the Hagen Balve Reef and Eifel region. Modern analogues of symbiont-bearing (light green squares) and symbiont-barren (dark green crosses) species from various locations were measured. A compilation of previously measured modern symbiont-bearing and symbiont-barren corals are also included from Swart (1983) and Frankowiak et al. (2016), where samples indicate distinct spaces for symbiont-bearing and symbiont-barren species as indicated by the respective field (adapted from Swart, 1983). Previously published Devonian values are shown as beige triangles, with darker and lighter brown tones indicating rugose and tabulate species, respectively. Notably, all Paleozoic samples from this study cluster within the symbiotic range and would indicate no difference between coral groups.

BIBLIOGRAPHY (I list here only papers not cited in the original contribution)

- Burchette, T. P. European Devonian reefs: a review of current concepts and models. *SEPM Spec. Public.* 30, 85–142 (1981).
- Hubert, B. L., Zapalski, M., Nicollin, J. P., Mistiaen, B., & Brice, D. (2007). Selected benthic faunas from the Devonian of the Ardennes: an estimation of palaeobiodiversity. *Acta Geologica Polonica*, 57(2), 223–262.
- McFadden CS, Quattrini AM, Brugler MR, Cowman PF et al. Phylogenomics, Origin, and Diversification of Anthozoans (Phylum Cnidaria). *Syst Biol.* 2021 Jun 16; 70(4):635–647 (2021).
- Lecompte M. J. (1939) Les tabulés du Dévonien moyen et supérieur du bord sud du bassin de Dinant. *Musée Royal D’histoire Naturelle De Belgique* 90, 1–229.
- Zapalski, M.K., Wrzolek, T., Skompski, S. et al. Deep in shadows, deep in time: the oldest mesophotic coral ecosystems from the Devonian of the Holy Cross Mountains (Poland). *Coral Reefs* 36, 847–860 (2017).
- Zapalski, M.K., Berkowski, B. The Silurian mesophotic coral ecosystems: 430 million years of photosymbiosis. *Coral Reefs* 38, 137–147 (2019).
- Mikolaj K. Zapalski

University of Warsaw

Referee #2 (Remarks to the Author):

This is a remarkable study that substantially increases our understanding of the origins of some of the most important symbiotic associations on Earth - those that underpin the immense productivity and success of coral reefs in deep time. The authors are amongst the best in the world at making painstaking measurements of the $\delta^{15}\text{N}$ of skeletal bound organic matter from fossil and modern coral specimens - through a clever method that has been pioneered and refined by several of the authors over the years. When those first methodological papers emerged it was obvious that the huge potential was to unlock an entirely new field of paleobiological reconstructions - not only for understanding past ocean processes in terms of chemical and physical paleoceanography - but now also understanding paleoecology.

It is critically important for science to piece together the origins of symbiosis - as we now understand how pervasive biological partnerships are in nature - even within our own human gut. For coral reefs, the symbiosis between dinoflagellate algae and their cnidarian hosts is the “poster-child” for mutualism and one that has a perilous future amidst the Anthropocene. Here, an elegant and meticulous methodology paired with extensive sampling of paleontological and modern collections creates a simple and compelling story that provides insights to where coral reefs come from, and helps us to use that historical context to see more clearly where they are going. The manuscript is of interest to a wide readership - including those interested in symbiosis, and bio geosciences.

As an isotopist myself I am best able to evaluate the $\delta^{15}\text{N}$ data component of this study and I find the results to be compelling. The manuscript clearly illustrates how the issue of organic contamination is mitigated. The remarkable consistency of pattern with low $\delta^{15}\text{N}$ in modern photosymbiotic corals similar to ancient colonial taxa is impressive. There is a robust signal - again consistent across deep time - seen in exclusively heterotrophic modern corals and their ancient counterparts. That the difference between symbiotic and aposymbiotic species hovers around 3‰ is also quite profound as this is around the generalization that around 3.4‰ is the average difference between consumers and their diet.

This paper thus has several key advances/outcomes that go well beyond refining the temporal origins of symbiosis in the sea.

Namely,

1. it confirms the high fidelity of biological recorders - in contrast to sedimentary records from mixed (abiotic and biotic) origins.
2. It firmly validates $\delta^{15}\text{N}$ as a strong indicator of photosymbiosis and the timing of its emergence as concomitant to widespread oceanic oligotrophy. The former is congruent with many studies of modern corals and the host-symbiont-skeletal compartment and the latter further helps us understand why mitigating local stressors like nutrient pollution is essential for having symbiotic corals.
3. It further establishes that other proxies such as oxygen and carbon isotopes are problematic with confounding diagenetic and kinetic isotope effects.
4. It paves the way for expanding the work through myriad applications in paleo research, including paleo-nutrient reconstructions on carbonate bound deposits on Earth.

5. I am not sure if this is a correct or impactful interpretation - but I found it interesting that the $\delta^{15}\text{N}$ of the reef matrix is more similar to that of the colonial taxa, suggesting that their bio calcification is the major driver of these deposits, as opposed to heterotrophic species.

The manuscript is very well written and clear. Most of the content is accessible to the general readership of the journal and the references therein are suitable for directing the readership to source materials where necessary. I have very few and very minor comments that do not detract from the paper at all, but could help improve it slightly.

We are grateful for the general comments and remarks of R2 and appreciate the detailed listing of the advances of the nitrogen isotope method for paleoecological investigations.

1. Symbiodiniaceae is misspelled. Line 55

We have addressed the misspelling.

2. Line 82 - best to avoid using zooxanthellae - an antiquated and outdated term.

We have removed the term.

3. The manuscript is largely written in the present tense - the past tense should be used particularly when presenting results that were previously measured.

We agree with R2 and have addressed that where applicable.

4. Lines 127-145 it would be informative to have sample sizes listed here with the means. It would also be good to understand why the variation for some species/locations is many times greater than other species/locations and whether or not that is an artifact of unbalanced replication.

We have included sample size for means (with $n=$ “ ”) and sample sizes are also listed in the supplementary tables.

5. Line 180 - as there is no direct evidence of the presence of symbionts by any corroborative means - the language here should be more deductive. That the patterns for modern symbiotic and aposymbiotic species are similar the authors deduce that the mechanism for this is the presence of photosymbionts that vector DIN to the holobiont.

We agree with R2, we have tried to use a more deductive language in the new version of the manuscript.

I whole-heartedly support publication.

David M. Baker

Referee #3 (Remarks to the Author):

I think this is an excellent study of the stable isotope paleoecology of a Devonian ecosystem but I don't think the Devonian data, as presented, paint a clear picture of photosymbiosis in corals. Accordingly I don't think this work in its current form rises to the level of publication in Nature.

The offset between rugose and tabulate corals is indistinguishable from a trophic effect with an offset of ~1 trophic level. In this case, the simplest argument is that the average difference of 2.86‰ is simply the result of a trophic isotope effect. This is an interesting paleoecological result but is not particularly strong evidence of photosymbiosis. Rugose corals were large animals and likely had prey that included larger organisms like zooplankton and maybe even small fish that were probably secondary consumers. In this case, it is likely that Rugose corals would have had d15N values that were higher than primary consumers. It is not that I think that tabulates didn't have photosymbionts, it's that I don't think your isotope data provide an unequivocal case for photosymbiosis in this environment.

We disagree with the interpretation of our data proposed by R3. R3 alludes to the possibility that the d15N difference between solitary/ceroid rugose corals and colonial tabulate/dendroid rugose corals from the same reef does not reflect the effects of N recycling by symbionts. Instead R3 suggests that this difference is driven by differences in the food sources of the corals.

First, we are not aware of any study on modern coral that supports the trophic niche partitioning suggested by R3. Although corals can occasionally consume zooplankton (or perhaps even small dying fish), available data indicate that these sources represent a negligible portion of their diet. For example,

In a study conducted by Price et al. 2023, coral d13C and d15N values of symbiont-bearing species plot on the lower-bound or even below that of POM (4-6‰) (Comm. Fig. 2). In contrast, lacking an effect of symbiosis on coral d15N, we would expect coral d15N values that are ~3‰ higher than their respective food source (suspended POM or zooplankton), yielding coral d15N of 9-13‰ in the Hawai'i system investigated in Comms. Fig. 2. far higher than observed in that study.

[REDACTED]

There is a growing body of observations in support of a direct effect of symbiosis on the d15N of corals. Studies of coral bleaching typically show a rise in the d15N of coral tissue (Erler et al., 2020; Rodrigues and Grottoli, 2006). Studies of facultatively symbiotic corals observe a lower d15N of the tissue in the symbiotic regions of a single coral colony (AGU 2024, OS22A-05). Isotopic measurements of multiple N forms, organisms, and corals across Bermuda are inconsistent with symbiotic corals having a trophic d15N elevation of ~3‰ relative to suspended POM or zooplankton, and are 2-3‰ lower in d15N than other ‘filter-feeders’ (serpulids) in the same reef (Luu, 2022).

In contrast, regarding the argument about the coral polyp/corallite size and their food source (corals with smaller polyps/corallites eating phytoplankton and corals with bigger polyps/corallites eating zooplankton or fish), we are not aware of any support for this interpretation. To the contrary, comparison of two small-polyp coral species (*Porites asteroides* and *Madracis decactis*) and two large-polyp coral species (*Diploria labyrinthiformis* and *Montastraea cavernosa*) from Bermuda reefs observe similar tissue and coral-bound d15N for the two groups (Luu, 2022).

Moreover, our data on rugose corals of different corallite sizes argue against the suggestion of R3 that the d15N differences can be explained solely by corallite size and trophic level. For example, *Mesophyllum*

secundum ($6.78 \pm 0.42\%$) and *Acanthophyllum concavum* ($7.69 \pm 0.12\%$) from West Sahara show similar $\delta^{15}\text{N}$ values compared to our *Disphyllum* species ($7.34 \pm 0.23\%$) despite having twice the corallite size (Comm. Fig. 2). On the other hand, *Dendrostella trigiemme* ($1.45 \pm 0.66\%$) showed similar $\delta^{15}\text{N}$ values compared to *Temnophyllum latum* ($5.16 \pm 0.88\%$) or *T. astricum* ($5.52 \pm 1.49\%$) (Comm Fig. 3). These findings contradict the hypothesis of R3, while also indicating that corallite size is not a definitive marker of photosymbiosis.

Comments Figure 3. Corallite sizes of colonial rugose corals with respect to their coral-bound $\delta^{15}\text{N}$ values from Sauerland and West Sahara. There is no consistent trend with corallite size and CB- $\delta^{15}\text{N}$ values.

You need to include ^{15}N values from the encasing sedimentary strata as opposed to the matrix. I agree that the matrix may be derived from crushed coral materials but you need the context of water column organic matter $\delta^{15}\text{N}$ values. There are many Devonian studies that rely on bulk ^{15}N and many other deep time sections from shallow sequences on continental crust and the general consensus is that the bulk values are probably pretty good reflections of primary processes (Robinson 2012 covers this issue nicely). Like black shales and sapropels

(e.g. Mediterranean) the bulk and compound specific values converge on ^{15}N depletion in these organic matter rich deposits and that the values are probably primary. The same logic applies for the Devonian. Sure, bulk values need to be viewed critically, but the body of work

out there suggests that they are probably pretty good approximations of the $\delta^{15}\text{N}$ of primary producers. And if we assume that the organic matter in the sediment is composed primarily of phytoplankton (a reasonable assumption) then the tabulates are probably just primary consumers and the rugose are secondary consumers. In any case you really need to report the bulk values and hash out why you don't think they are any good or not.

Measurements of the uncleaned bulk sedimentary matrix should be equivalent to the classical bulk sedimentary measurements that are conducted in an Elemental Analyzer. The studies on bulk sedimentary $\delta^{15}\text{N}$ in Devonian sediments mentioned by R3 are from areas of high organic matter content (Total N content in the $\mu\text{mol} - \text{mmol} / \text{mg}$ range) (Algeo et al., 2014; Mercuzot et al., 2021; Percival et al., 2019). In this study, we report the first Devonian values of bulk sedimentary $\delta^{15}\text{N}$ in low-organic content reef sediments (max. 7.2 nmol per gram of sediment in unclean samples). The N content in our samples is 3-5 orders of magnitude lower than those reported in previous studies of high organic-rich shales. Therefore, these $\delta^{15}\text{N}$ data are from very different environments that cannot be compared.

Our data show that uncleaned bulk sedimentary $\delta^{15}\text{N}$ values in these low-N environments are not suitable for reconstructing changes in the Devonian N cycle or environmental baselines. Studies in the modern ocean show that in these low-organic content sediments, the effect of diagenesis on bulk $\delta^{15}\text{N}$ can be very large (up to 3-6‰) (Altabet and Francois, 1994; Robinson et al., 2012). In addition, comparison of fossil-bound $\delta^{15}\text{N}$ with bulk sediment $\delta^{15}\text{N}$ from the Quaternary and throughout the Cenozoic indicate that bulk sedimentary $\delta^{15}\text{N}$ measurements can be heavily altered by diagenesis and/or terrestrial inputs in a variety of depositional environments, including: the subantarctic Southern Ocean (Martinez-Garcia et al., 2014; Robinson et al., 2005), the South China Sea (Ren et al 2017 PNAS), the Caribbean Sea (Ren et al., 2009; Straub et al., 2013), and the North Pacific (Kast et al., 2019; Ren et al., 2015). For example, in the Caribbean bulk sedimentary $\delta^{15}\text{N}$ shows nearly constant values across the last glacial-interglacial cycle while two separate foraminifera species show foraminifera-bound $\delta^{15}\text{N}$ records with strong precessional changes (Comms. Fig. 4a; Straub et al. 2013). A similar situation is observed in the South China Sea, where different species of foraminifera yield coherent, orbitally structured $\delta^{15}\text{N}$ record, while bulk sediment $\delta^{15}\text{N}$ from different sedimentary environments in the South China Sea yield vastly different records (Ren et al., 2017b). Studies through the Cenozoic show that bulk measurements underestimate the environmental signals recorded by foraminifera (Kast et al., 2019). In addition, recent comparisons of foraminifera-bound $\delta^{15}\text{N}$ and bulk sedimentary $\delta^{15}\text{N}$ have also shown that bulk sedimentary $\delta^{15}\text{N}$ can be altered even in high depositional environments (Comms. Fig. 4b; Studer et al., 2021).

[REDACTED]

In contrast, so long as the diagenetic conditions are appropriate to preserve the biomineral in question, fossil-bound d15N appears to be remarkably robust. For example, laboratory experiments have shown that repeated chemical oxidations, calcite dissolution, and heating (up to 300-400°C) does not affect the d15N of the organic matter that is protected in the mineral matrix. In addition, a number of studies have shown that the N content of fossils of the same species/genus remains relatively stable across thousands to millions of years, suggesting that mineral matrix acted as a closed system with respect to N (Kast et al., 2019; Leichliter et al., 2021; Ren et al., 2017a). Furthermore, foraminifera, corals and diatoms, which have different sensitivities to alteration, provide consistent estimates of regional d15N changes when measured across the same time periods and in the same regions (e.g., Wang et al., 2017)

In oceanic regions of complete nitrate consumption, coral-bound d15N values have been shown to be reliable recorders of the d15N of the nitrate supplied to the euphotic zone, especially in low-organic environments with complete nitrate consumption (Comm. Fig. 5) (Wang et al., 2016, 2015).

[REDACTED]

Our new data from the eastern Rheic Ocean shows shifts in the baseline that are consistently captured by pairs of symbiont-bearing and -barren coral species, so that the difference between them remains constant. In contrast, bulk sedimentary ^{15}N measurements show a high degree of variability across basins (Figure. 2 and 3 in main text).

What is the source of the N in the spar and why does it have the same ^{15}N of the corals?

We have measured clean and unclean secondary sparite and find that the d^{15}N of cleaned sparite reflects coral-bound d^{15}N more closely. As we carefully try to avoid any sparite or encasing sedimentary matrix we do not provide any explicit explanation on why that might be the case. However, some of our internally conducted laboratory experiments allude to the fact that partial dissolution of carbonates does not alter the pristine d^{15}N signal. As sparite is recrystallized carbonate that usually builds up in submarine environments, our dissolution experiments could indicate that partial dissolution removes

small amounts of organic matter from the most abundant skeletal material that could be trapped by re-precipitation processes

(i.e., sparite) (Martínez-García et al., 2022).

I know that maybe abstracts don't count as the "first" of some kind of analysis, but there is a AGU abstract from 2017 (Hickey) on Devonian coral nitrogen that comes up when you search on "coral nitrogen isotopes Devonian" in google scholar. It has not been published in any other form as far as I can tell. I also don't think that being first to do something is very interesting and you should probably remove reference to it? Lots of firsts are not better than some piece of science that came afterwards. If the results are in a highly cited journal like Nature, it seems sort of self-evident that it is new.

We have also not found any evidence that the given data of the AGU abstract have ever been published. In any case, this would be the first peer-reviewed $\delta^{15}\text{N}$ measurements in Paleozoic corals.

The conodont temperatures at Hagen on the low end are typical for rocks of this age, but on the high end, that is quite high. In both cases we are talking about rocks that are well past the oil window. Is there any concern that the organic matter in the corals and their pore spaces is

derived from migrating oil?

We are not aware of any descriptions of migrating oil in any of our locations. The extremely low N content of the uncleaned samples is inconsistent with this scenario. In any case, migrating oil could only have the possibility to contaminate unclean bulk sediment. As we limit our interpretations to the crystal-bound organic matter and use extensive cleaning protocols, even migrating oil could not affect organic material inherent to the coral. If that was the case, we would see significant differences in samples that are below (i.e., Eifel region) and above (i.e., Hagen Balve reef) the oil window, which is not the case.

The methods are very detailed and the lengths to which the authors have gone to work with these difficult, nitrogen lean materials is impressive.

Some questions regarding methodological details:

It seems you have done a good job cleaning your samples and the methods are well thought out to be sure you have clean coral carbonate, very nice indeed. Though I am curious about the interiors of the corals. Rugose corals are quite porous and well-preserved specimens can retain the original voids and pores. Are these all filled with spar?

We appreciate the comments of R3. We address the issue of potential sparite contamination by carefully drilling samples layer-by-layer. In our ‘worst-case’ scenario regarding diagenetic temperature (max 300°C at Hagen Balve reef), we have analyzed clean and unclean sparite as well as clean and unclean rugose coral samples and found pure unclean sparite to show high average d15N values (~5‰) while it converged to lower d15N values in cleaned sparite samples (Extended Data Fig. 4 in main text). If we saw significant contamination from sparite in our rugose samples, we would see higher d15N values in uncleaned rugose samples and lower d15N in cleaned rugose samples. However, we see the exact opposite pattern.

Does the rather higher thermal maturity make the persulfate oxidation difficult? Are you sure that the kerogen or other organic phases are fully oxidized? Can you determine if the nitrogen and carbon content of the cleaned skeleton match up to the yielded N content in the dissolved sample?

As we dissolve carbonate samples with HCl prior to oxidation, thermal maturity does not matter in the oxidation step. We have shown that in our thermal alteration experiments,

whereby samples heated to 500°C were fully dissolved and oxidized to yield sufficient N for the injection to our bacteria (Martínez-García et al., 2022).

Why do you think that the unclean sedimentary matrix has such incredibly broad d15N values? I have never seen anything like this. A few permil perhaps, but the range is startling.

Again, the fundamental difference with respect to previous study is the concentration of N in this samples (3-6 orders of magnitude lower). These low concentrations could make diagenetic effect more evident, but it will also make the samples more prone to contamination during storage and sampling. This combination of factors could explain the broad range of values found in our samples. These results stress the importance of careful cleaning protocols.

Were you specifically worried about clay in your drilled samples? I am curious as to why you were concerned about the clay if you are milling out what was hopefully primary carbonate.

We use a clay-removal as a precautionary step to remove any potentially left-over contamination, this method has been validated in previous studies measuring foraminifera and deep sea corals (Martinez-Garcia et al., 2014; Ren et al., 2009, 2017a; Robinson et al., 2012).

Why are you concerned about oxides?

Metal oxides are a concern because they could trap organic material during precipitation. The long diagenetic history of each setting made us decide to use the strongest cleaning protocols to rule out any contamination that we know could bias pristine $\delta^{15}\text{N}$ values.

Did you measure the ^{15}N of the sparite or any secondary cements? This may be useful to determine if secondary calcification had any nitrogen from outside sources.

We have analyzed cleaned and uncleaned secondary sparite from the location (Hagen Balve Reef) with the highest diagenetic temperature based on the Conodont Alteration Index (Extended Data Fig. 4). The differences with the corals are minimal suggesting that there was no incorporation of N from external sources. These results are consistent with measurements in recrystallized Cenozoic foraminifera, which show no signs of N isotopic alteration (Kast et al., 2019).

Sources

- Algeo, T.J., Meyers, P.A., Robinson, R.S., Rowe, H., Jiang, G.Q., 2014. Icehouse–greenhouse variations in marine denitrification. *Biogeosciences* 11, 1273–1295. <https://doi.org/10.5194/bg-11-1273-2014>
- Altabet, M.A., Francois, R., 1994. The Use of Nitrogen Isotopic Ratio for Reconstruction of Past Changes in Surface Ocean Nutrient Utilization, in: Zahn, R., Pedersen, T.F., Kaminski, M.A., Labeyrie, L. (Eds.), *Carbon Cycling in the Glacial Ocean: Constraints on the Ocean’s Role in Global Change*, NATO ASI Series. Springer, Berlin, Heidelberg, pp. 281–306. https://doi.org/10.1007/978-3-642-78737-9_12
- Coates, A.G., Jackson, J.B.C., 1987. Clonal growth, algal symbiosis, and reef formation by corals. *Paleobiology* 13, 363–378. <https://doi.org/10.1017/S0094837300008988>
- Dworczak, P.G., Correa, M.L., Jakubowicz, M., Munnecke, A., Joachimski, M.M., Mazzoli, C., Berkowski, B., 2022. Carbon and oxygen isotope fractionation in the Late Devonian heterocoral Oligophylloides: Implications for the skeletogenesis and evolution of the Heterocorallia. *Palaeogeography, Palaeoclimatology, Palaeoecology* 598, 111017. <https://doi.org/10.1016/j.palaeo.2022.111017>
- Erler, D.V., Rangel, M.S., Tagliafico, A., Riekenberg, J., Farid, H.T., Christidis, L., Scheffers, S.R., Lough, J.M., 2020. Can coral skeletal-bound nitrogen isotopes be used as a proxy for past bleaching? *Biogeochemistry* 151, 31–41. <https://doi.org/10.1007/s10533-020-00706-2>
- Frankowiak, K., Wang, X.T., Sigman, D.M., Gothmann, A.M., Kitahara, M.V., Mazur, M., Meibom, A., Stolarski, J., 2016. Photosymbiosis and the expansion of shallow-water corals. *Sci. Adv.* 2, e1601122. <https://doi.org/10.1126/sciadv.1601122>
- Jakubowicz, M., Berkowski, B., Correa, M.L., Jarochovska, E., Joachimski, M., Belka, Z., 2015. Stable Isotope Signatures of Middle Palaeozoic Ahermatypic Rugose Corals – Deciphering Secondary Alteration, Vital Fractionation Effects, and Palaeoecological Implications. *PLOS ONE* 10, e0136289. <https://doi.org/10.1371/journal.pone.0136289>
- Kast, E.R., Stolper, D.A., Auderset, A., Higgins, J.A., Ren, H., Wang, X.T., Martínez-García, A., Haug, G.H., Sigman, D.M., 2019. Nitrogen isotope evidence for expanded ocean suboxia in the early Cenozoic. *Science* 364, 386–389. <https://doi.org/10.1126/science.aau5784>
- Leichliter, J.N., Lüdecke, T., Foreman, A.D., Duprey, N.N., Winkler, D.E., Kast, E.R., Vonhof, H., Sigman, D.M., Haug, G.H., Clauss, M., Tütken, T., Martínez-García, A., 2021. Nitrogen isotopes in tooth enamel record diet and trophic level enrichment: Results from a controlled feeding experiment. *Chemical Geology* 563, 120047. <https://doi.org/10.1016/j.chemgeo.2020.120047>
- Luu, V.H.-Y., 2022. Modern-ocean ground-truthing of nitrogen isotopes in scleractinian corals: A proxy for surface ocean nutrient conditions.
- Martínez-García, A., Jung, J., Ai, X.E., Sigman, D.M., Auderset, A., Duprey, N.N., Foreman, A., Fripiat, F., Leichliter, J., Lüdecke, T., Moretti, S., Wald, T., 2022. Laboratory Assessment of the Impact of Chemical Oxidation, Mineral Dissolution, and Heating on the Nitrogen Isotopic Composition of Fossil-Bound Organic Matter. *Geochemistry, Geophysics, Geosystems* 23, e2022GC010396. <https://doi.org/10.1029/2022GC010396>
- Martínez-García, A., Sigman, D.M., Ren, H., Anderson, R.F., Straub, M., Hodell, D.A., Jaccard, S.L., Eglinton, T.I., Haug, G.H., 2014. Iron Fertilization of the Subantarctic Ocean During the Last Ice Age. *Science* 343, 1347–1350. <https://doi.org/10.1126/science.1246848>
- Mercuzot, M., Thomazo, C., Schnyder, J., Pellenard, P., Baudin, F., Pierson-Wickmann, A.-C., Sans-Jofre, P., Bourquin, S., Beccalotto, L., Santoni, A.-L., Gand, G., Buisson, M., Glé, L., Munier, T., Saloume, A., Boussaid, M., Boucher, T., 2021. Carbon and Nitrogen Cycle Dynamic in Continental Late-Carboniferous to Early Permian Basins of Eastern Pangea (Northeastern Massif Central, France). *Frontiers in Earth Science* 9.
- Percival, L.M.E., Selby, D., Bond, D.P.G., Rakociński, M., Racki, G., Marynowski, L., Adatte, T., Spangenberg, J.E., Föllmi, K.B., 2019. Pulses of enhanced continental weathering associated with multiple Late Devonian climate perturbations: Evidence from osmium-isotope compositions.

Palaeogeography, Palaeoclimatology, Palaeoecology 524, 240–249.
<https://doi.org/10.1016/j.palaeo.2019.03.036>

Price, J.T., McLachlan, R.H., Jury, C.P., Toonen, R.J., Grottoli, A.G., 2021. Isotopic approaches to estimating the contribution of heterotrophic sources to Hawaiian corals. *Limnology and Oceanography* 66, 2393–2407. <https://doi.org/10.1002/lno.11760>

Ren, H., Chen, Y.-C., Wang, X.T., Wong, G.T.F., Cohen, A.L., DeCarlo, T.M., Weigand, M.A., Mii, H.-S., Sigman, D.M., 2017a. 21st-century rise in anthropogenic nitrogen deposition on a remote coral reef. *Science* 356, 749–752. <https://doi.org/10.1126/science.aal3869>

Ren, H., Sigman, D.M., Martínez-García, A., Anderson, R.F., Chen, M.-T., Ravelo, A.C., Straub, M., Wong, G.T.F., Haug, G.H., 2017b. Impact of glacial/interglacial sea level change on the ocean nitrogen cycle. *Proc Natl Acad Sci USA* 114, E6759–E6766. <https://doi.org/10.1073/pnas.1701315114>

Ren, H., Sigman, D.M., Meckler, A.N., Plessen, B., Robinson, R.S., Rosenthal, Y., Haug, G.H., 2009. Foraminiferal Isotope Evidence of Reduced Nitrogen Fixation in the Ice Age Atlantic Ocean. *Science* 323, 244–248. <https://doi.org/10.1126/science.1165787>

Ren, H., Studer, A.S., Serno, S., Sigman, D.M., Winckler, G., Anderson, R.F., Oleynik, S., Gersonde, R., Haug, G.H., 2015. Glacial-to-interglacial changes in nitrate supply and consumption in the subarctic North Pacific from microfossil-bound N isotopes at two trophic levels. *Paleoceanography* 30, 1217–1232. <https://doi.org/10.1002/2014PA002765>

Robinson, R.S., Kienast, M., Albuquerque, A.L., Altabet, M., Contreras, S., Holz, R.D.P., Dubois, N., Francois, R., Galbraith, E., Hsu, T.-C., Ivanochko, T., Jaccard, S., Kao, S.-J., Kiefer, T., Kienast, S., Lehmann, M., Martinez, P., McCarthy, M., Möbius, J., Pedersen, T., Quan, T.M., Ryabenko, E., Schmittner, A., Schneider, R., Schneider-Mor, A., Shigemitsu, M., Sinclair, D., Somes, C., Studer, A., Thunell, R., Yang, J.-Y., 2012. A review of nitrogen isotopic alteration in marine sediments. *Paleoceanography* 27. <https://doi.org/10.1029/2012PA002321>

Robinson, R.S., Sigman, D.M., DiFiore, P.J., Rohde, M.M., Mashiotta, T.A., Lea, D.W., 2005. Diatom-bound $^{15}\text{N}/^{14}\text{N}$: New support for enhanced nutrient consumption in the ice age subantarctic. *Paleoceanography* 20. <https://doi.org/10.1029/2004PA001114>

Rodrigues, L.J., Grottoli, A.G., 2006. Calcification rate and the stable carbon, oxygen, and nitrogen isotopes in the skeleton, host tissue, and zooxanthellae of bleached and recovering Hawaiian corals. *Geochimica et Cosmochimica Acta* 70, 2781–2789. <https://doi.org/10.1016/j.gca.2006.02.014>

Straub, M., Sigman, D.M., Ren, H., Martínez-García, A., Meckler, A.N., Hain, M.P., Haug, G.H., 2013. Changes in North Atlantic nitrogen fixation controlled by ocean circulation. *Nature* 501, 200–203. <https://doi.org/10.1038/nature12397>

Studer, A.S., Mekik, F., Ren, H., Hain, M.P., Oleynik, S., Martínez-García, A., Haug, G.H., Sigman, D.M., 2021. Ice Age-Holocene Similarity of Foraminifera-Bound Nitrogen Isotope Ratios in the Eastern Equatorial Pacific. *Paleoceanography and Paleoclimatology* 36, e2020PA004063. <https://doi.org/10.1029/2020PA004063>

Swart, P.K., 1983. Carbon and oxygen isotope fractionation in scleractinian corals: a review. *Earth-Science Reviews* 19, 51–80. [https://doi.org/10.1016/0012-8252\(83\)90076-4](https://doi.org/10.1016/0012-8252(83)90076-4)

Wang, X.T., Sigman, D.M., Cohen, A.L., Sinclair, D.J., Sherrell, R.M., Cobb, K.M., Erler, D.V., Stolarski, J., Kitahara, M.V., Ren, H., 2016. Influence of open ocean nitrogen supply on the skeletal $\delta^{15}\text{N}$ of modern shallow-water scleractinian corals. *Earth and Planetary Science Letters* 441, 125–132. <https://doi.org/10.1016/j.epsl.2016.02.032>

Wang, X.T., Sigman, D.M., Cohen, A.L., Sinclair, D.J., Sherrell, R.M., Weigand, M.A., Erler, D.V., Ren, H., 2015. Isotopic composition of skeleton-bound organic nitrogen in reef-building symbiotic corals: A new method and proxy evaluation at Bermuda. *Geochimica et Cosmochimica Acta* 148, 179–190. <https://doi.org/10.1016/j.gca.2014.09.017>

Wang, X.T., Sigman, D.M., Prokopenko, M.G., Adkins, J.F., Robinson, L.F., Hines, S.K., Chai, J., Studer, A.S., Martínez-García, A., Chen, T., Haug, G.H., 2017. Deep-sea coral evidence for lower Southern Ocean surface nitrate concentrations during the last ice age. *Proceedings of the National Academy of Sciences* 114, 3352–3357. <https://doi.org/10.1073/pnas.1615718114>

Reviewer Reports on the First Revision:

Referees' comments:

Referee #1 (Remarks to the Author):

The paper by Jung et al. "Coral Photosymbiosis on Mid-Devonian Reefs" presents results on the studies of nitrogen stable isotopes of Palaeozoic coral skeletons (representatives of Rugosa and Tabulata, comparatively with Scleractinia) in relation to photosymbiosis. Brief characteristics of the work have been included in my previous review, so I will comment rather on changes done by the authors.

I am glad that authors were able to sample more corals, especially the reef-forming alveolitids and pachyporids. This makes the work much more representative for the tabulate corals as a group and, consequently, much more representative for the Palaeozoic reefs. It is striking that tabulates are generally characterized by very similar N isotopic composition across the sites, and in my opinion, this is a very strong point of the work. Still, however, several ecologically important groups (e. g. platy or frondescent mesophotic tabulates) are not sampled (as suggested in my previous review). It is a pity, as such isotopic analyses of the groups that have the strongest morphological evidence of photosymbiosis would be probably very interesting and useful for other studies.

I have several minor comments that could be addressed in the revised version of the work:

1. At the beginning (line 18) authors state "It is currently unclear whether photosymbiosis first arose in the Triassic, with the emergence of scleractinian corals, or if it was already prevalent amongst older coral groups that have since gone extinct^{3–5}". The youngest of the three cited works is 26 years old, and the oldest 37, and many works have been published since then (and are cited within the manuscript); when it comes to certain groups of tabulates (small-polyped alveolitids or platy coenitids) there is a general agreement that these must have been photosymbiotic (e.g., Bridge et al. 2022). I suggest rephrasing this sentence.

2. Authors state (line 85 and onwards) "As a result morphology alone appears insufficient to distinguish between photosymbiont-bearing and -barren coral species". This problem somewhat more complicated than the statement suggests: for example large sizes of polyps may occur in both photosymbiont-bearing and photosymbiont-barren taxa, very tiny polyp diameters are characteristic for aposymbiotic taxa. Therefore, while in certain taxa the morphological features may be ambiguous, in others they may clearly point towards photosymbiosis. Another example of the same issue may be exemplified on the colony integration – low integration may occur in all groups, but high colony integration is known only from photosymbiotic taxa. What is more, there is a very recent work by Krol et al. (2024; <https://doi.org/10.1007/s00338-023-02450-z>) that addresses the morphological criteria related to photosymbiosis.

3. The corals added to the analysis after the review are not illustrated, so it is difficult to assess the correctness of the taxonomic identification. I would encourage authors to add the photographs of the thin sections to the supplementary materials.

4. Line 84. plural should be “genera”

5. Lines 97-98 authors are mixing colony form and colony structure, which refer to two distinct anatomical features

6. l. 97-98 I am not sure what the authors mean by “loosely colonial (dendroid, cerioid)”. While dendroid corals (such as *Tubastraea*) are often pseudocolonial, cerioidal structure is a typical, colonial structure of moderate integration.

7. Check the spelling of “Moroccan” throughout the text (not “Moroccon”).

8. In some places (e.g., line 143) authors refer to “*Thamnopora urensis*”, whereas in others (Suppl. Table 2) to “*Th. urensis germanica*”. Please, unify the naming of taxa and check them throughout the text.

9. Line 257 – “suggest”

10. Line 282 – “indicate”

Despite the lack of coverage of all ecologically important coral groups I think that after the changes and additional analyses the manuscript has been significantly improved.

Referee #2 (Remarks to the Author):

I remain very confident that this is an exceptional scientific contribution to the understanding of the origins of symbiosis in the oceans. I am satisfied with the reviewer’s detailed responses to all reviews.

Response to Reviews

Referee #1 (Remarks to the Author):

The paper by Jung et al. “Coral Photosymbiosis on Mid-Devonian Reefs” presents results on the studies of nitrogen stable isotopes of Palaeozoic coral skeletons (representatives of Rugosa and Tabulata, comparatively with Scleractinia) in relation to photosymbiosis. Brief characteristics of the work have been included in my previous review, so I will comment rather on changes done by the authors.

I am glad that authors were able to sample more corals, especially the reef-forming alveolitids and pachyporids. This makes the work much more representative for the tabulate corals as a group and, consequently, much more representative for the Palaeozoic reefs. It is striking that tabulates are generally characterized by very similar N isotopic composition across the sites, and in my opinion, this is a very strong point of the work. Still, however, several ecologically important groups (e. g. platy or frondescent mesophotic tabulates) are not sampled (as suggested in my previous review). It is a pity, as such isotopic analyses of the groups that have the strongest morphological evidence of photosymbiosis would be probably very interesting and useful for other studies.

We appreciate the supportive nature of R1’s comments. In the future, we hope to analyze some of the recommended mesophotic tabulates. However, our study focusses on shallow, oligotrophic marine environments; mesophotic reefs are beyond the scope of this study. We are looking forward to explore more open questions in upcoming research.

1. At the beginning (line 18) authors state “It is currently unclear whether photosymbiosis first arose in the Triassic, with the emergence of scleractinian corals, or if it was already prevalent amongst older coral groups that have since gone extinct^{3–5}”. The youngest of the three cited works is 26 years old, and the oldest 37, and many works have been published since then (and are cited within the manuscript); when it comes to certain groups of tabulates (small-polyped alveolitids or platy coenitids) there is a general agreement that these must have been photosymbiotic (e.g., Bridge et al. 2022). I suggest rephrasing this sentence.

We have changed the sentence to “The evolutionary history of this symbiosis might clarify its organismal and environmental roles³, but its prevalence through time and across taxa, morphologies, and oceanic settings is currently unclear^{4–6}” and have included Bridge et al. 2022 as reference 6.

2. Authors state (line 85 and onwards) “As a result morphology alone appears insufficient to distinguish between photosymbiont-bearing and -barren coral species”. This problem somewhat more complicated than the statement suggests: for example large sizes of polyps may occur in both photosymbiont-bearing and photosymbiont-barren taxa, very tiny polyp diameters are characteristic for aposymbiotic taxa. Therefore, while in certain taxa the morphological features may be ambiguous, in others they may clearly point towards photosymbiosis. Another example of the same issue may be exemplified on the colony

integration – low integration may occur in all groups, but high colony integration is known only from photosymbiotic taxa. What is more, there is a very recent work by Krol et al. (2024) that addresses the morphological criteria related to photosymbiosis.

We have changed the sentence to “As a result, morphological features alone cannot conclusively identify symbiosis across all taxa.” We added Król et al. 2024 as reference 51 to this sentence, based on the following statement from that paper: “The matter of photosymbiosis in Paleozoic corals remains unresolved as it is not possible to directly check for the presence of algal symbionts in fossil corals.”

While we recognize the progress made with morphological analysis, we need to point out that a number of corals identified by our study as photosymbiotic would not have been considered as candidates based on identified morphological characteristics that we find to be symbiotic (e.g., *Dendrostella trigemme*, a dendroid rugose coral with low colony integration). Conversely, we are able to rule out symbiosis in species with higher colony integration that would be considered, based on morphological characteristics, to be candidates for symbiosis (e.g., *Argutastraea quadrigemina*).

We agree with R1 that morphology, such as platy growth forms, point to the presence of photosymbionts in Paleozoic corals. However, morphology is generally a function of various abiotic and biotic factors, such as life history, depth distribution, wave action or turbidity. R1 writes in one of his papers that “There are two possible functions of platy colony morphologies: stabilization in soft sediment and photoadaptive growth” (Zapalski et al. 2019) and, thus, that studies on Paleozoic morphology require more careful considerations of the substratum as well. While some Paleozoic corals might exhibit comparable morphological traits to modern symbiotic corals, it remains ambiguous to distinguish between all possible symbiont-bearing and -barren species on Paleozoic reefs. For example, there are modern coral species (i.e., *Oculina arbuscula* or *Astrangia poculata*) that exhibit facultative symbiosis, have high colony integration and a depth range up to 200m (Rivera and Davies, 2021). Symbiotic versus aposymbiotic specimens of these species could not be distinguished by morphological analysis, as their morphology would remain the same. In contrast, we have analyzed a specimen of *Oculina arbuscula* from a shallow reef (8m depth) in Jamaica and found it to be dominantly asymbiotic.

3. The corals added to the analysis after the review are not illustrated, so it is difficult to assess the correctness of the taxonomic identification. I would encourage authors to add the photographs of the thin sections to the supplementary materials.

We added images of all species that were analyzed in the supplements (Supplementary Fig. 1 and 2). In our analysis we focused on material that was taxonomically identified by Dr. Rudolf Birenheide who curated the collection at Senckenberg in Frankfurt am Main back at the time and was one of the most respected Paleozoic coral taxonomists.

4. Line 84. plural should be “genera”

We have adjusted the plural to genera

5. Lines 97-98 authors are mixing colony form and colony structure, which refer to two distinct anatomical features

We have adjusted the sentence so that the colony structure is in parentheses.

6. 1. 97-98 I am not sure what the authors mean by “loosely colonial (dendroid, cerioid)”. While dendroid corals (such as *Tubastraea*) are often pseudocolonial, cerioid structure is a typical, colonial structure of moderate integration.

We have adjusted the wording and allude to the colonial sub-types (or structure) we have analyzed. The sentence is updated to “We focused mainly on tabulate corals, colonial (dendroid, phaceloid, ceroid) and solitary rugose corals,…”

7. Check the spelling of “Moroccan” throughout the text (not “Moroccon”).

We have changed the spelling throughout the manuscript.

8. In some places (e.g., line 143) authors refer to “*Thamnopora urens*”, whereas in others (Suppl. Table 2) to “*Th. urens germanica*”. Please, unify the naming of taxa and check them throughout the text.

We appreciate the comment and have adjusted the taxonomic description to “*Thamnopora urens*” throughout the text.

9. Line 257 – “suggest”

We have adjusted the spelling.

10. Line 282 – “indicate”

We have adjusted the spelling.

Despite the lack of coverage of all ecologically important coral groups I think that after the changes and additional analyses the manuscript has been significantly improved.

We thank R1 for this supportive comment.

Referee #2 (Remarks to the Author):

I remain very confident that this is an exceptional scientific contribution to the understanding of the origins of symbiosis in the oceans. I am satisfied with the reviewer's detailed responses to all reviews.

We thank R2 for the input throughout the review process.

References

- Bridge, T.C.L., Baird, A.H., Pandolfi, J.M., McWilliam, M.J., Zapalski, M.K., 2022. Functional consequences of Palaeozoic reef collapse. *Sci Rep* 12, 1386. <https://doi.org/10.1038/s41598-022-05154-6>
- Rivera, H.E., Davies, S.W., 2021. Symbiosis maintenance in the facultative coral, *Oculina arbuscula*, relies on nitrogen cycling, cell cycle modulation, and immunity. *Sci Rep* 11, 21226. <https://doi.org/10.1038/s41598-021-00697-6>
- Król, J. J., Berkowski, B., Denayer, J. & Zapalski, M. K. Deducing photosymbiosis in extinct heliolitid corals. *Coral Reefs* **43**, 91–105 (2024).

Reviewer Reports on the Second Revision:

Referees' comments:

Referee #2 (Remarks to the Author):

I reiterate my strong support for the publication of this work. The other reviewer has provided detailed feedback which has been adequately addressed.